# Cell-type-specific responses to associative learning in the primary motor cortex

Candice Lee[1], Emerson F Harkin[1], Xuming Yin[1], Richard Naud[1,2,3,4], Simon Chen[1,3,4]*

[1]Department of Cellular and Molecular Medicine, University of Ottawa, Ottawa, Canada; [2]Department of Physics, STEM Complex, University of Ottawa, Ottawa, Canada; [3]Brain and Mind Research Institute, University of Ottawa, Ottawa, Canada; [4]Center for Neural Dynamics, University of Ottawa, Ottawa, Canada

**Abstract** The primary motor cortex (M1) is known to be a critical site for movement initiation and motor learning. Surprisingly, it has also been shown to possess reward-related activity, presumably to facilitate reward-based learning of new movements. However, whether reward-related signals are represented among different cell types in M1, and whether their response properties change after cue–reward conditioning remains unclear. Here, we performed longitudinal in vivo two-photon $Ca^{2+}$ imaging to monitor the activity of different neuronal cell types in M1 while mice engaged in a classical conditioning task. Our results demonstrate that most of the major neuronal cell types in M1 showed robust but differential responses to both the conditioned cue stimulus (CS) and reward, and their response properties undergo cell-type-specific modifications after associative learning. PV-INs' responses became more reliable to the CS, while VIP-INs' responses became more reliable to reward. Pyramidal neurons only showed robust responses to novel reward, and they habituated to it after associative learning. Lastly, SOM-INs' responses emerged and became more reliable to both the CS and reward after conditioning. These observations suggest that cue- and reward-related signals are preferentially represented among different neuronal cell types in M1, and the distinct modifications they undergo during associative learning could be essential in triggering different aspects of local circuit reorganization in M1 during reward-based motor skill learning.

*For correspondence:
schen2@uottawa.ca

Competing interest: The authors declare that no competing interests exist.

## Editor's evaluation

Using advanced live brain imaging techniques, the authors studied the activities of neurons in the primary motor cortex of mice during a classical conditional task, in which a tone is paired with a water reward. They found that distinct types of neurons respond differently to the auditory cue or the reward, and the responses evolve differentially as learning proceeds. This work reveals an interesting role of the motor cortex beyond its well-recognized function in motor control and suggests distinct functions of pyramidal neurons as well as various interneurons in reinforcement learning.

## Introduction

The primary motor cortex (M1) is an essential site for movement execution and motor learning. Within M1, neurons encode movement goals and movement kinematics (*Georgopoulos et al., 1992*; *Moran and Schwartz, 1999*; *Peters et al., 2014*). Intriguingly, neurons in M1 have also been reported to show reward-related activity. In vivo recording studies performed in nonhuman primates found neurons in M1 that encode reward anticipation, reward delivery, and mismatches between the two (*Marsh et al., 2015*; *Ramakrishnan et al., 2017*; *Ramkumar et al., 2016*). In human subjects, reward has also been shown to modulate M1 activity, likely through an inhibitory circuit-dependent mechanism (*Thabit*

*et al., 2011*). However, it remains unclear how reward-related responses are represented in M1, and if the representation changes with associative learning.

It was recently shown that in well-trained mice performing a skilled reaching task, a subset of layer 2/3 (L2/3) pyramidal neurons (PNs) in M1 specifically report successful, but not failed, reach-and-grasp movements. In contrast, a different subset of PNs report only failed reach-and-grasp movements (*Levy et al., 2020*). Since the ability to use past experience to learn action–outcome associations is critical to survival, encoding the outcome in M1 may be an important part of motor skill learning. It is widely accepted that associative learning using reinforcement can accelerate and enhance learning (*Abe et al., 2011*; *Nikooyan and Ahmed, 2015*). In the case of motor learning, studies have demonstrated that positive feedback (reward) facilitates motor memory retention and negative feedback (punishment) speeds up the learning process (*Galea et al., 2015*). One hypothesis is that during learning, reward signals in the brain, together with neuromodulators and synaptic plasticity, are involved in potentiating and optimizing the neural circuitry in M1 that underlies the rewarded movement. Implementing such a learning process would necessitate the interplay between different cell types within the local microcircuitry (*Richards et al., 2019*).

M1, like other cortical areas, is densely packed with PNs and diverse inhibitory interneuron (IN) types and is wired in a delicately balanced and intricate circuit. Different IN subtypes have been shown to have distinct gene expression profiles, electrophysiological properties, and connectivity motifs (*Fishell and Rudy, 2011*; *Markram et al., 2004*). Somatostatin-, parvalbumin-, and vasoactive intestinal peptide- expressing inhibitory neurons (SOM-INs, PV-INs, and VIP-INs, respectively) are three major nonoverlapping subtypes of GABAergic neurons that broadly form a common microcircuit motif in the cortex. Some studies have demonstrated that SOM-INs preferentially target distal dendrites of PNs to filter synaptic inputs, fast-spiking PV-INs preferentially target perisomatic regions of PNs enabling strong inhibition of spiking, and VIP-INs regulate local microcircuits by controlling other local INs (*Pfeffer et al., 2013*). Due to their diverse properties and strategic connectivity motifs, these INs exert fine control over local network activity and provide a potential mechanism for how the brain processes reward signals and ultimately uses this information to optimize neural activity related to learned motor skills.

Multiple studies using in vivo opto-recordings in the primary visual cortex have shown that visual orientation selectivity in PNs is modulated and sharpened by PV- and SOM-INs (*Atallah et al., 2012*; *Lee et al., 2012*; *Wilson et al., 2012*). In the primary auditory cortex, PV- and SOM-INs exert analogous control over PN frequency tuning (*Seybold et al., 2015*). Moreover, in the auditory cortex, prefrontal cortex, and basolateral amygdala, reinforcement signals such as reward and punishment have been shown to recruit VIP-INs, which in turn, inhibit SOM- and PV-INs (*Krabbe et al., 2019*; *Pi et al., 2013*). The subtype-specific roles of these INs have long been elusive, but a complex picture is emerging where INs are not only responsible for maintaining a delicate balance of excitation and inhibition, but are also actively involved in processing activity in the cortex (*Lee et al., 2020*; *Wood et al., 2017*).

Here, we employed chronic in vivo two-photon imaging, combined with a head-fixed classical conditioning task, to monitor the activity of the same population of PNs, PV-INs, SOM-INs, or VIP-INs before and after associative learning to investigate whether and how conditioned cue stimulus (CS) and unconditioned reward are represented among different neuronal cell types in M1. Our results demonstrate that all four major cell types in M1 show distinct responses to CS and reward, and their response properties undergo cell-type-specific modifications after associative learning. Notably, PV-INs and VIP-INs exhibited stimulus-specific modifications, in which PV-INs became more reliably responsive to the CS but not to the reward, whereas VIP-INs became more reliable to the reward but not to the CS. PNs initially showed robust responses to novel reward but became habituated to it after associative learning. Lastly, SOM-IN responses emerged with learning and responded more reliably to both the CS and reward. Taken together, these results show that cue- and reward-related signals are preferentially represented among major neuronal cell types in M1, and they undergo cell-type-specific modifications during associative learning, indicating they may have distinct roles in integrating reinforcement signals to promote circuit reorganization in M1 during motor skill learning.

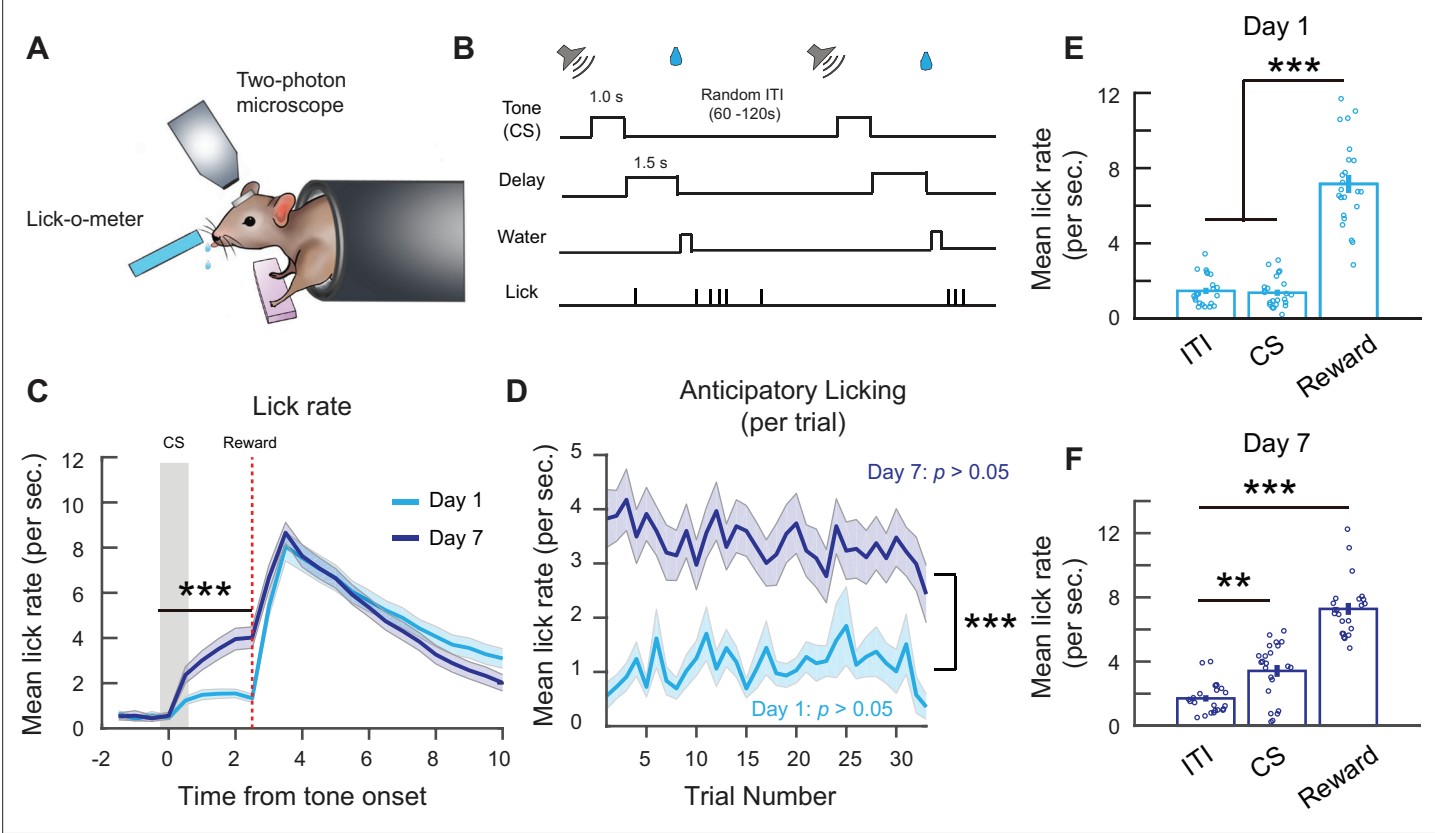

**Figure 1.** Associative learning during a head-fixed classical conditioning task. (**A**) Schematic of head-fixed classical conditioning task. (**B**) Trial structure. (**C**) Mean lick rate per second on days 1 and 7. Binned over 0.5-s intervals. Lick rate following cue stimulus (CS) onset up to reward delivery time is higher on day 7. Two-way analysis of variance (ANOVA), ***$p < 1 \times 10^{-3}$, effect of time: $p < 1 \times 10^{-3}$, effect of day: $p < 1 \times 10^{-3}$. (**D**) Mean anticipatory lick rate across trials within days 1 and 7 sessions. Mean anticipatory lick rate was calculated from CS onset to end of delay period. Two-way ANOVA, effect of trial number: $p = 0.91$, effect of day: $p < 1 \times 10^{-3}$. (**E**) Mean lick rate during the first 2.5 s of intertrial interval (ITI) lick bouts, 2.5 s following CS onset and 2.5 s following reward delivery on day 1. Each point is the mean from an individual mouse. One-way ANOVA with Tukey–Kramer correction for multiple comparisons, ITI vs. CS: $p = 0.97$, ITI vs. reward: $p < 1 \times 10^{-3}$, CS vs. reward: $p < 1 \times 10^{-3}$. (**F**) Mean lick rate during the first 2.5 s of ITI lick bouts, 2.5 s following CS onset and 2.5 s following reward delivery on day 7. Each point is the mean from an individual mouse. One-way ANOVA with Tukey–Kramer correction for multiple comparisons. ITI vs. CS: $p = 1.08 \times 10^{-3}$, ITI vs. reward: $p < 1 \times 10^{-3}$. $n = 23$ mice. **$p < 0.01$, ***$p < 0.001$. Error bars show standard error of the mean (SEM).

## Results

To understand how reward-associated signals are represented within the local microcircuitry in M1 before and after associative learning, we established a head-fixed auditory cued reward conditioning task, which allowed us to combine the task with in vivo two-photon Ca²⁺ imaging to examine the response properties of different neuronal cell-type populations in awake and behaving mice (*Figure 1A*). In this task, water-restricted mice were exposed to a conditioned stimulus (CS; auditory tone, 1-s duration), followed by a 1.5-s delay and then the delivery of the unconditioned stimulus (US; water reward, ~10 µl). Mice were trained for ~30–35 trials/session (1 session/day for 7 days) with randomly varied intertrial intervals (ITIs) between 60 and 120 s (*Figure 1B*). Since M1 is known to be involved in movement initiation and motor skill learning, we chose to use a simple classical conditioning task with just an auditory tone paired with reward and omitted any additional training where mice would be required to learn a new movement. The rationale for this is that many neuronal cell types, including PNs, PV-, and SOM-INs, have been shown to undergo modifications when mice acquire new movements (*Chen et al., 2015*; *Cichon and Gan, 2015*; *Donato et al., 2013*; *Xu et al., 2009*). However, since licking is an innate movement that does not induce plastic changes in adult mice (*Chen et al., 2015*; *Komiyama et al., 2010*; *Peters et al., 2014*), we can reliably attribute

changes in neuronal activity over the course of the task to associative learning, rather than motor learning.

Mice learned to associate the CS with the reward after 7 days, shown by an increase in anticipatory lick rate, a conditioned response, following the cue onset on day 7 compared to day 1 (*Figure 1C*). On a trial-by-trial basis, anticipatory lick rate did not change significantly within a single session on both days 1 and 7, implying limited within-session improvements (*Figure 1D*). To ensure the increase in lick rate was specific to the CS, we compared the mean lick rate during the CS and reward period to the lick rate during self-initiated spontaneous lick bouts in the ITI (in the absence of the CS or reward). To be consistent with the 2.5-s analysis window for CS responses, we analyzed the first 2.5 s of ITI lick bouts and 2.5 s following reward delivery. On day 1, the mean lick rate during ITI lick bouts (1.47 ± 0.17/s) and the CS (1.37 ± 0.17/s) were similar, while the lick rate following reward delivery was significantly higher (7.16 ± 0.48/s). In contrast, on day 7 following associative learning, the lick rate during the CS period (3.42 ± 0.37/s) was significantly higher than during ITI lick bouts (1.71 ± 0.2/s), demonstrating that the mice effectively learned the CS–reward association by day 7 (*Figure 1E, F*).

To investigate the activity of different neuronal cell types during this task, we used in vivo two-photon $Ca^{2+}$ imaging of different cell-type populations. To target PNs in M1, we injected an adeno-associated virus (AAV) carrying a $Ca^{2+}$ indicator (GCaMP6f) driven by the CaMKII promoter (AAV1. CaMKIIa.GCaMP6f) into M1 of wild-type B6129S mice. After 3–5 weeks, we recorded the activity of hundreds of L2/3 PNs using two-photon microscopy in awake mice while they underwent the head-fixed conditioning task, and we tracked the same population of neurons on days 1 and 7 (*Figure 2A*). We identified all the active neurons within a session, irrespective of the behavioral task (see Methods), and sorted neurons by the timing of their peak activity relative to the CS onset. It was apparent that there were subpopulations of neurons more responsive to CS, reward, or both (*Figure 2B, C*). We also repeated the experiments to examine if the major IN subtypes in M1 also respond to the CS and reward during the conditioning task. To do this, we injected AAV-Syn-Flex-GCaMP6f in PV-Cre, SOM-Cre, or VIP-Cre transgenic mice to selectively express GCaMP6f in PV-INs, SOM-INs, or VIP-INs, respectively, and then performed in vivo two-photon $Ca^{2+}$ imaging to monitor the response properties of the same population of INs on days 1 and 7 after associating learning (*Figure 2—figure supplements 1 and 2*). We compared the mean percentage of active cells within the entire session to ensure all cell types had a similar proportion of active cells (irrespective of the behavioral task) on days 1 and 7 (*Figure 2D*).

To examine task-related activity in each cell type, we first compared the mean percent of active cells during the CS and reward to a null distribution made by randomly sampling the session irrespective of the behavioral task, and then calculating the mean percentage of active neurons during the sampled period. By repeating this 1000 times for each cell type on days 1 and 7, we created a distribution of the percentage of active neurons that were present at baseline levels or by chance. Surprisingly, we found that only PV-IN and VIP-IN cell types had a percent of CS- and reward-responsive cells that were significantly greater than chance level on both days 1 and 7 (PV-IN CS: day 1: 15.26% ± 2.11%, day 7: 24.09% ± 2.98%; PV-IN reward: day 1: 20.17% ± 3.27%, day 7: 27.47% ± 3.66%; VIP-IN CS: day 1: 11.29% ± 3.23%, day 7: 16.59% ± 2.01%, VIP-IN reward: day 1: 18.65% ± 5.91%, day 7: 26.3% ± 6.04%; *Figure 2F, G*). PN responses to the CS were not different from the null distribution on both days 1 and 7 (day 1: 18.42% ± 1.05%, day 7: 16.41% ± 1.33%; *Figure 2E*); in contrast, PN responses to the reward were significantly higher on day 1 but significantly lower than the null distribution on day 7 (day 1: 23.95% ± 2.49%, day 7: 12.12% ± 0.95%; *Figure 2E*). Lastly, SOM-INs showed significant responses to the CS and reward only on day 7 following associative learning, while on day 1, they demonstrated no response to the CS and a modest response to the reward (CS: day 1: 5.53% ± 2.7%, day 7: 13.56% ± 3.17%; Reward: day 1: 9% ± 3.66%, day 7: 12.15% ± 3.93%; *Figure 2H*). Based on these findings, we decided to only examine the sessions where the percent of active cells were significantly greater than the null distribution in our subsequent analysis, as a nonsignificant percent of active cells during the stimulus period cannot be readily distinguished from nontask-related baseline noise.

We began our analysis on PV-INs and VIP-INs because they both showed significant responses to both the CS and reward on days 1 and 7. To understand how their representations of reward and reward-associated cues changed over the course of learning, we first analyzed the tuning of individual cells to unbiasedly identify their response properties. By quantifying the tuning of each cell's

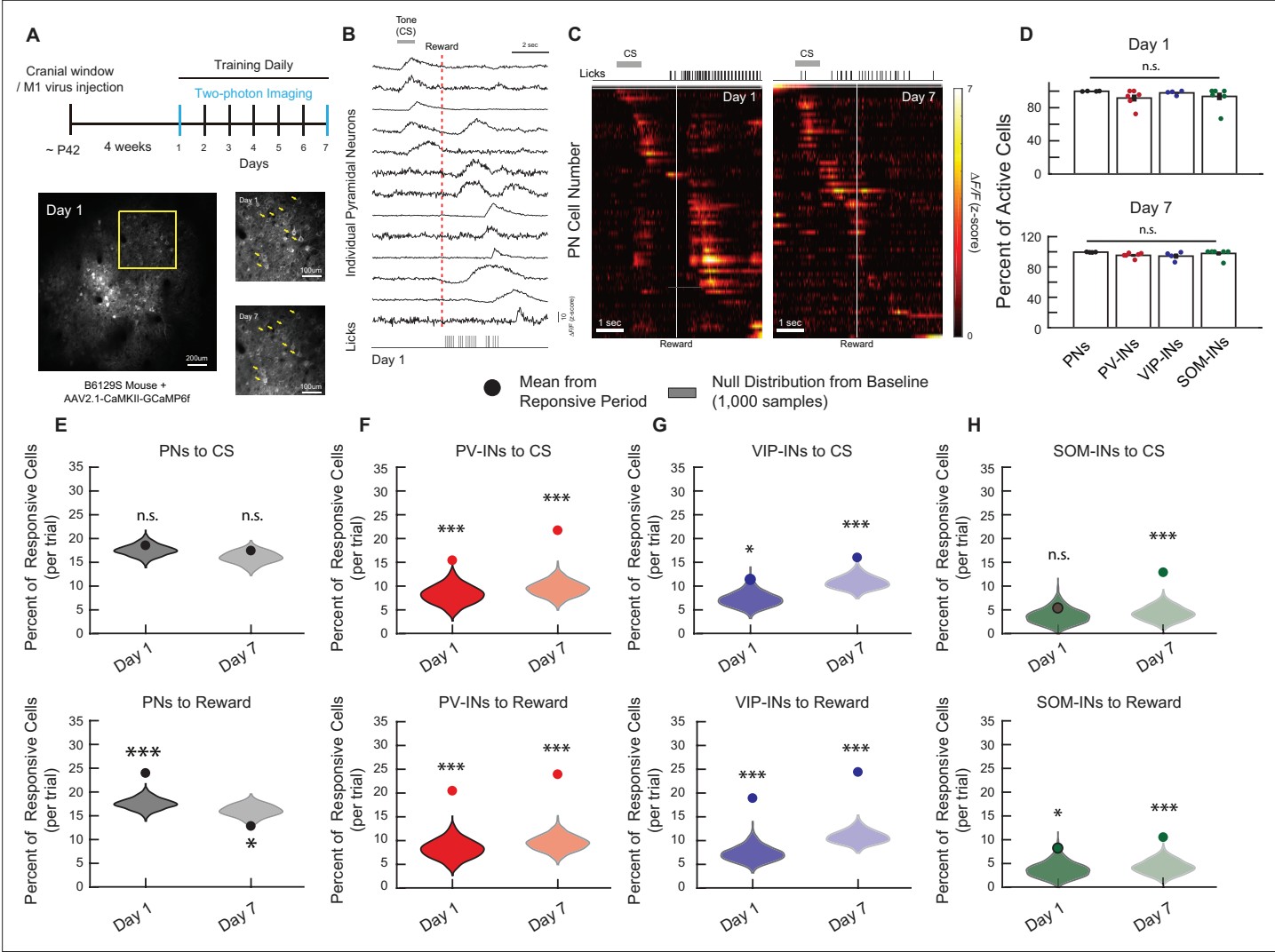

**Figure 2.** Longitudinal in vivo Ca²⁺ imaging of neuronal responses in M1 during a classical conditioning task. (**A**) Experimental timeline (top). In vivo two-photon imaging of L2/3 pyramidal neurons (PNs) expressing GCaMP6f in M1 (bottom left). The same population can be tracked from days 1 to 7 (bottom right). Yellow arrows indicate example tracked neurons across days. (**B**) Z-Scored fluorescence traces from 13 neurons (top), and the corresponding licking measured with the lick-o-meter (bottom) from the same mouse and same trial on day 1. Gray bar represents the timing of the cue stimulus (CS). Dotted red line indicates the onset of water reward delivery. (**C**) Z-Scored activity of all the active neurons from an example mouse during one representative trial on days 1 and 7, sorted by timing of maximum activity following the CS onset. Gray bar represents the timing of the CS. White line indicates the onset of water reward delivery. (**D**) Mean percent of active neurons within a session, irrespective of the behavioral task, for PNs, PV-INs, VIP-INs, and SOM-INs on days 1 and 7. All cell types showed a similar percentage of active neurons. One-way analysis of variance (ANOVA), n.s., nonsignificant, day 1: p = 0.38, day 7: p = 0.22. Error bars show standard error of the mean (SEM). (**E–H**) Mean percent of responsive neurons to CS (top) and reward (bottom) within 2.5 s of CS/reward onset for each cell type. Violin plots show null distribution of percentage of responsive neurons made by randomly resampling mice and shuffling the session, 1000 times (see Methods). The circle represents the mean percentage of tone- or reward-responsive neurons. Monte-Carlo with Bonferroni correction, n.s., nonsignificant, *p < 0.05, ***p < 1 × 10⁻³. PN CS day 1: p = 0.582, CS day 7: p = 0.423, Reward day 1: p < 1 × 10⁻³, Reward day 7: p = 0.015, n = 1029 cells from six mice (**E**), PV-IN CS day 1: p < 1 × 10⁻³, CS day 7: p < 1 × 10⁻³, Reward day 1: p < 1 × 10⁻³, Reward day 7: p < 1 × 10⁻³, n = 316 cells from six mice (**F**), VIP-IN CS day 1: p = 0.039, CS day 7: p < 1 × 10⁻³, Reward day 1: p < 1 × 10⁻³, Reward day 7: p < 1 × 10⁻³, n = 407 cells from four mice (**G**), SOM-IN CS day 1: p = 0.47 , CS day 7: p < 1 × 10⁻³, Reward day 1: p = 0.033, Reward day 7: p < 1 × 10⁻³, n = 189 cells from seven mice (**H**).

The online version of this article includes the following figure supplement(s) for figure 2:

**Figure supplement 1.** Z-Scored ΔF traces from cells among each cell type on days 1 and 7.

**Figure supplement 2.** Z-Scored population activity of cells tracked from days 1 to 7 for each cell type.

average response during the CS and reward response periods (2.5-s window) using the nonparametric Spearman correlation $\rho$ (see Methods), we observed a wide range of tuning coefficients to the CS and reward, with a small proportion that was strongly positively or negatively tuned to the CS or reward stimulus (tuning coefficient near –1 or 1; *Figure 3A–D*), consistent with our earlier analyses demonstrating that neurons in M1 show activity associated with the CS or reward during the conditioning task. We next examined whether the tuning coefficient within each cell type changed after associative learning by calculating the change in tuning coefficients for each cell between days 1 and 7. Again, to validate our findings, we compared these values to a null distribution of $\Delta\rho$ values obtained by randomly sampling the two sessions (see Method details). The PV-IN population did not show any significant changes in either CS or reward tuning between days 1 and 7 ($\bar{\Delta}\rho_{CS} = -0.049 \pm 0.046$, $\bar{\Delta}\rho_{reward} = 0.014 \pm 0.054$; *Figure 3E, F*), indicating that neither CS- nor reward-related tuning became stronger after associative learning. In contrast, VIP-INs' CS tuning did not change significantly between days 1 and 7 $\bar{\Delta}\rho_{CS} = -0.065 \pm 0.048$, but VIP-INs' reward tuning significantly increased on day 7 ($\bar{\Delta}\rho_{reward} = 0.161 \pm 0.086$; *Figure 3G, H*), suggesting a strengthening of VIP-IN responsivity to reward following associative learning.

Although the tuning properties can reveal changes in task-related responsivity, this analysis is limited in identifying changes at the trial-by-trial level. When we assessed population activity following the CS onset (*Figure 4A*), it was apparent that a group of PV-INs and VIP-INs were responsive to CS on both days 1 and 7 (*Figures 4B and 5B*). Hence, by identifying and tracking the same neurons from days 1 to 7, we were able to ask if there was (1) an increase in the number of neurons being recruited as CS or reward responsive during associative learning or (2) a change in the trial-by-trial reliability of CS and reward responses. When we compared the mean percent of CS-responsive neurons on days 1 and 7, we found that the average percent of CS-responsive PV-INs during a trial increased significantly by day 7 (day 1: 15.26% ± 2.11%, day 7: 24.09% ± 2.98%; *Figure 4C*), while the percent of CS-responsive VIP-INs did not change (day 1: 11.29% ± 3.23%, day 7: 16.59% ± 2.01%; *Figure 4D*), demonstrating that more PV-INs became responsive to the CS after associative learning. We then assessed the reliability of the responses, defined as the percent of trials within a session where a neuron was responsive to the CS. This measure quantifies how consistently a neuron responded to the CS within a session. We first plotted the cumulative distribution function of reliabilities among all PV-INs and VIP-INs. We observed that PV-INs, as a population, were significantly more reliable in their CS responses than VIP-INs on day 1 (*Figure 4E*). When we sorted neurons based on their day 1 reliability values and followed them to day 7, we observed that many of the PV-INs that initially had Low Reliability to CS became more responsive on day 7 (*Figure 4H*); therefore, we grouped neurons into 'High Reliability' if they were among the top 50th percentile, while neurons in the lower 50th percentile were deemed 'Low Reliability'. We found that PV-INs that began as highly reliable maintained their reliability to the CS (day 1: 29.8% ± 1.51%, day 7: 33.87% ± 4.72%), while PV-INs that began as Low Reliability became significantly more reliable (8.47% ± 0.46%, day 7: 18.99% ± 3.76%; *Figure 4F*). In contrast, the reliability of both High and Low VIP-INs did not change (High Reliability: day 1: 26.55% ± 2.62%, day 7: 25.93% ± 3.81%; Low Reliability: day 1: 6.32% ± 0.76%, day 7: 14.24% ± 2.6%; *Figure 4G*). We then followed individual Low Reliability PV-INs and calculated the change in reliability to the CS (*reliability$_{CS}$* from days 1 to 7. As a control, we randomly sampled the day 7 session irrespective of the behavioral task and calculated a reliability value. We then subtracted that value from the actual day 1 CS reliability to generate a randomized change in reliability (*reliability$_{random}$* for each Low Reliability neuron (see Methods). When we compared the two distributions, we found that among the Low Reliability PV-INs, *reliability$_{CS}$* was significantly greater than the *reliability$_{random}$* control (*Figure 4I*). Lastly, we examined if the onset of neuronal activity after the CS changed after associative learning, and if any neurons that were previously responsive to the CS became responsive to the reward only. We did not observe a change in the onset of neuronal activity following CS (*Figure 4—figure supplement 1A*, see Methods). Furthermore, 97% of the CS-responsive PV-INs from day 1 still showed responsivity to CS on day 7 (*Figure 4—figure supplement 1C, D*). Altogether, these results show that as a population, more PV-INs became responsive to the CS, and this is mainly due to Low Reliability PV-INs that became more reliable following associative learning.

To demonstrate that changes in PV-INs' representation of the CS resulted from associative learning, we conducted additional control experiments to examine their responses to tone when mice received no rewards or nonpaired randomly timed rewards. In the first experiment, water-restricted PV-Cre

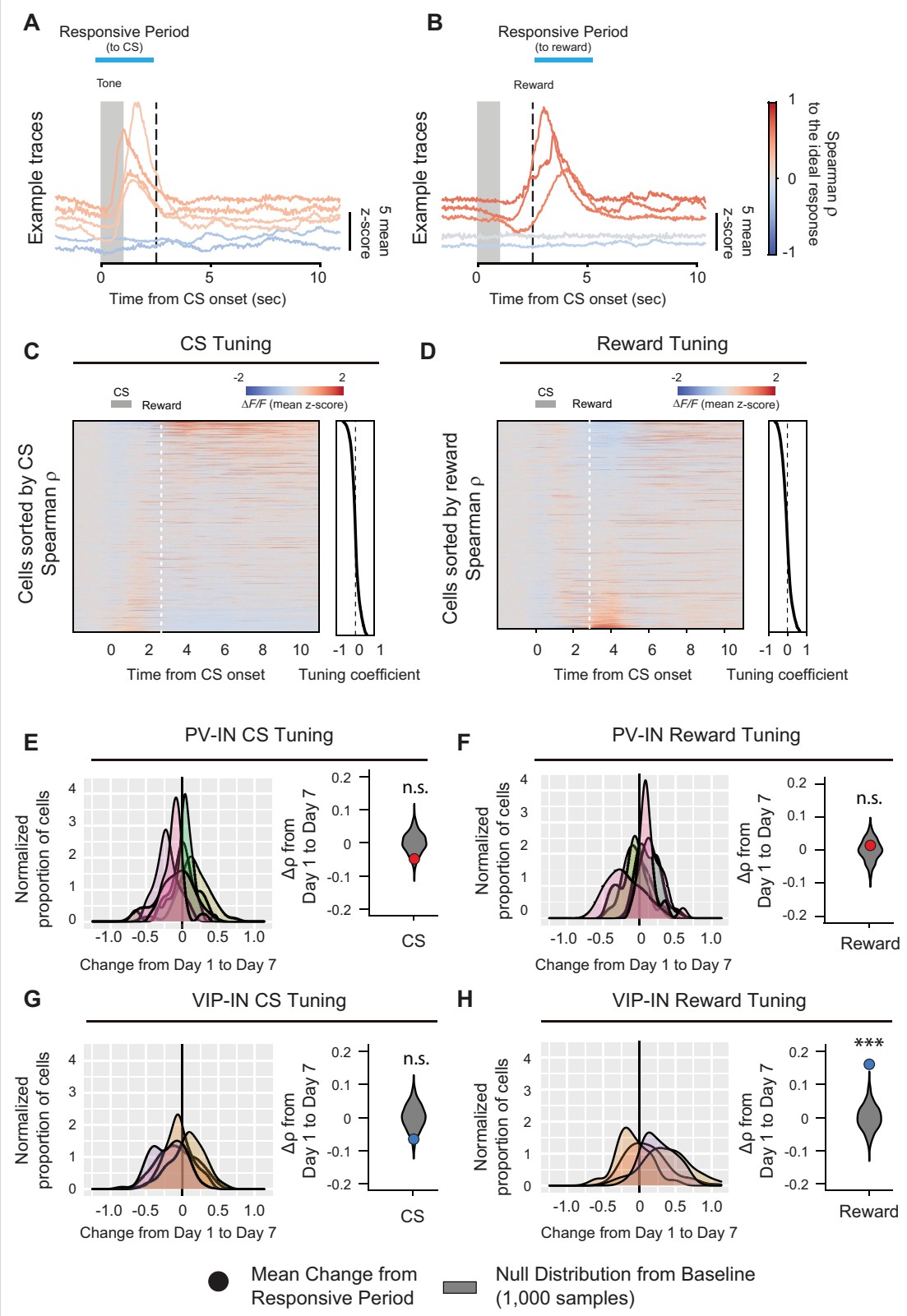

**Figure 3.** Learning-associated changes in single-neuron tuning properties in M1. Example fluorescence traces, color-coded based on nonparametric Spearman correlation with cue stimulus (CS) (**A**) or reward (**B**). Each trace is a trial-averaged response from different neurons on day 7. Trial-averaged fluorescence of all the neurons recorded on day 7 and sorted to the value of the Spearman correlation (−1–1) to CS (**C**) or reward (**D**). Active neurons during the CS- or reward-responsive period showed higher tuning coefficient. Left, distribution of changes in Spearman correlation Δρ with CS (**E, G**)

*Figure 3 continued on next page*

*Figure 3 continued*

or reward (**F, H**) for PV-INs (top) and VIP-INs (bottom). Each curve represents a Gaussian kernel density estimate of the distribution of $\Delta\rho$ in a single mouse. Right, mean change in Spearman correlation $\overline{\Delta\rho}$ for PV-INs and VIP-INs. Null distributions (gray) were estimated by resampling each mouse and shuffling trials 1000 times (see description of calculation of tuning coefficients in Methods). VIP-IN reward tuning significantly increased with associative learning. Monte-Carlo, ***p < 1 × 10⁻³, n.s., nonsignificant, PV-IN CS p = 0.12 (**E**), PV-IN Reward p = 0.61 (**F**), VIP-IN CS p = 0.082 (**G**), VIP-IN Reward p < 1 × 10⁻³ (**H**) PV-IN: *n* = 316 cells from six mice. VIP-IN: *n* = 407 cells from four mice.

mice were exposed to the same auditory tone used as the CS (1-s duration) but all water rewards were omitted (randomly varied ITI between 60 and 120 s; ~30 trials/session; 1 session/day for 7 days). We imaged PV-IN activity on both days 1 and 7 and assessed their population responses following the tone onset (*Figure 4—figure supplement 2A*). We first compared the mean percent of active cells during tone to a null distribution made by randomly sampling the session irrespective of the behavioral task (as in *Figure 2*). Surprisingly, PV-INs did not show significant tone-responsive cells compared to the chance level on either day 1 or 7 (*Figure 4—figure supplement 2B*); hence, the average percent of tone-responsive PV-INs per trial also did not increase from days 1 to 7 (*Figure 4—figure supplement 2C*). Next, in a separate cohort of mice, we exposed water-restricted PV-Cre mice to tone, followed by a 'nonpaired' water reward that was given at randomly varied time intervals (40–80 s). Mice were also trained for ~30 trials/session (1 session/day for 7 days) with a randomly varied ITI between 15 and 25 s (*Figure 4—figure supplement 2D*). We found PV-INs were significantly responsive to the tone stimuli on day 1, similar to what we observed earlier in the CS–reward task (*Figure 2F*). Interestingly, by day 7, PV-INs no longer responded to the tone stimulus (*Figure 4—figure supplement 2G*). We next examined if mice that received the tone stimulus with nonpaired water reward learned to associate the two after 7 days. We found the animals did not learn the association, as their conditioned response (tone-evoked anticipatory licking) did not increase at day 7 (*Figure 4—figure supplement 2E, F*). Unlike the mice that learned the association in the CS–reward task, we did not observe a change in the mean percent of tone-responsive PV-INs from days 1 to 7 in the nonpaired paradigm, and the reliability of Low Reliability PV-INs to tone also did not change (*Figure 4—figure supplement 2H, I*). Together, these results suggest that PV-INs in M1 do not respond to auditory tone in general, but instead only respond to the tone when the animal actively associates it with reward. Moreover, the changes among PV-INs to the CS tone from days 1 to 7 are specific to associative learning.

We next assessed reward responses among PV-INs and VIP-INs in the same manner but now looked for responses within 2.5 s of the reward delivery time (*Figures 4B and 5A, B*). We tracked the same neurons from days 1 to 7 and compared the mean percent of reward-responsive neurons. PV-INs and VIP-INs did not show a significant change in the percent of responsive cells per trial (PV-IN: day 1: 20.17% ± 3.27%, day 7: 27.47% ± 3.66%; VIP-IN: day 1: 18.65% ± 5.91%, day 7: 26.3% ± 6.04%; *Figure 5C, D*). When we examined the cumulative distribution of reliabilities for reward responses between the two cell types, VIP-INs as a population were significantly more reliable than PV-INs on day 1 (*Figure 5E*). By dividing the cells into High and Low Reliability groups, we found the High Reliability VIP-INs maintained their reliability (VIP-IN High Reliability: day 1: 38.79% ± 2.71%, day 7: 35.88% ± 4.32%), and the Low Reliability VIP-INs became significantly more reliable on day 7 (VIP-IN Low Reliability: day 1: 10.25% ± 1.67%, day 7: 24.06% ± 4.87%; *Figure 5G, H*). In contrast, both High and Low Reliability PV-INs maintained their reliability to reward (High Reliability: day 1: 35.59% ± 2.81%, day 7: 41.94% ± 5.7%; Low Reliability: day 1: 10.58% ± 0.74%, day 7: 19.59% ± 3.08%; *Figure 5F*). We also followed individual Low Reliability VIP-INs and calculated the change in reliability to reward (*reliability$_{reward}$*) from days 1 to 7. As a control, we randomly sampled the day 7 session irrespective of the behavioral task and calculated a random *reliability$_{random}$* for each neuron as described above. When we compared the two distributions, the *reliability$_{reward}$* was significantly greater than the *reliability$_{random}$*, demonstrating that Low Reliability VIP-INs became more reliably responsive to reward (*Figure 5I*). We also examined if the onset of neuronal activity after reward consumption changed after associative learning, and if neurons that were previously responsive to reward became responsive to the CS only. We did not observe a change in the onset of neuronal activity following reward (*Figure 4—figure supplement 1B*). Furthermore, 95% of the VIP-INs from day 1 still showed responsivity to reward on day 7 (*Figure 4—figure supplement 1E, F*). Lastly, to demonstrate that changes in VIP-INs' representation of reward resulted from associative learning, we examined both PV-IN and VIP-IN responses to reward in mice that were exposed to the nonpaired behavioral paradigm (tone+

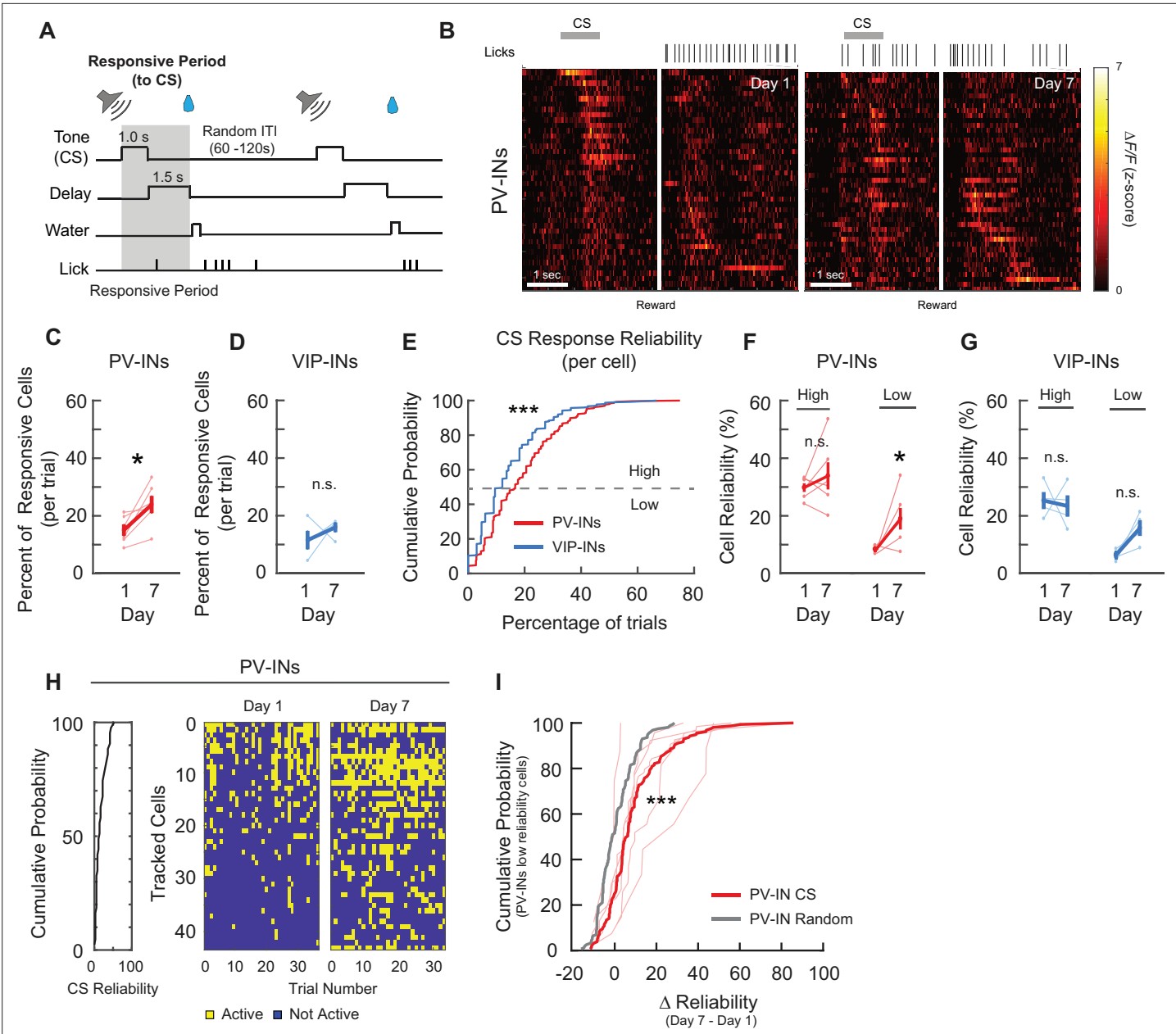

**Figure 4.** PV-IN and VIP-IN cue stimulus (CS)-related responses before and after associative learning. (**A**) Trial structure. Gray shaded bar represents the response period analyzed for CS-responsive activity. (**B**) Z-Scored activity of all the active PV-INs from an example mouse during one representative trial on days 1 and 7, sorted by timing of maximum activity following the CS onset. Gray bar represents the timing of the CS. White line indicates the onset of water reward delivery. Mean percent of cells that are responsive to the CS for PV-INs (**C**) and VIP-INs (**D**). PV-INs showed an increase in the percent of CS-responsive neurons after reinforcement learning, while VIP-INs did not show any change. Paired *t*-test, *p < 0.05, n.s., nonsignificant, PV-IN: p = 0.031 (**C**), VIP-IN: p = 0.38 (**D**). (**E**) Cumulative probability plots showing the percent of trials that each neuron responded to the CS for PV-INs and VIP-INs on day 1. Neurons from each cell type were pooled across mice. PV-INs showed significantly greater reliability to the CS than VIP-INs. Kolmogorov–Smirnov test, p < 1 × 10⁻³. Mean reliability index of cells that are responsive to the CS for PV-INs (**F**) and VIP-INs (**G**). Each cell type is divided into High or Low Reliability Group based on the 50th percentile from the cumulative probability plots in (**E**). High Reliability PV-INs maintained their consistency, while the Low Reliability group became more consistent in their responses to CS. (**G**) VIP-INs did not show a change in either group after learning. Paired *t*-test, PV-IN High Reliability: p = 0.38, PV-IN Low Reliability: p = 0.044, VIP-IN High Reliability: p = 0.79, VIP-IN Low Reliability: p = 0.060. (**H**) CS responses from all tracked PV-INs in one example mouse on days 1 and 7. Left, cumulative distribution of CS response reliability among all tracked cells within the example mouse. Right, binary map of each cell's CS response (active or not) across all trials on days 1 and 7. Cells were sorted by their day 1 reliability shown on the left and the order is maintained on day 7. Cells with low response reliability to CS on day 1 became more reliable on day 7. (**I**) Cumulative probability plots of the change in reliability from days 1 to 7 (*reliability_CS*) among PV-INs with Low Reliability group to CS on day 1.

*Figure 4 continued on next page*

**Figure 4 continued**

Bold red, *reliability*$_{CS}$ of the population of PV-INs. Thin red lines show the *reliability*$_{CS}$ distribution within individual PV-Cre mice. As a control, day 7 session was randomly sampled and a random reliability was calculated. *reliability*$_{random}$ was calculated by subtracting day 1 CS reliability from the day 7 random reliability (gray, *reliability*$_{random}$ from the same population of PV-INs). Kolmogorov–Smirnov test, ***p < $1 \times 10^{-3}$ PV-IN: $n$ = 316 cells from six mice. VIP-IN: $n$ = 407 cells from four mice. Error bars show standard error of the mean (SEM).

The online version of this article includes the following figure supplement(s) for figure 4:

**Figure supplement 1.** Response properties of PV-IN and VIP-IN cells tracked from days 1 to 7 are consistent across days.

**Figure supplement 2.** PV-IN and VIP-IN population plasticity is specific to associative learning.

randomly timed water rewards; *Figure 4—figure supplement 2D–F*). Consistent to what we observed in the CS–reward paradigm, both PV-INs and VIP-INs consistently showed higher mean percent of active cells during reward (2.5 s from the first lick after reward delivery) compared to the chance level on both days 1 and 7 (*Figure 4—figure supplement 2J–M*). However, because these mice did not learn the association between the auditory tone and randomly delivered water reward, we did not see an increase in the reliability of the Low Reliability VIP-INs from days 1 to 7 (*Figure 4—figure supplement 2O*), or a change in the percent of responsive cells in either PV-INs or VIP-INs (*Figure 4—figure supplement 2K–N*). Altogether, we found that during associative learning, while the proportion of reward-responsive VIP-INs during a given trial did not change, a subset of VIP-INs that were largely unresponsive to reward on day 1 became more reliably responsive on day 7.

Although PV-INs and VIP-INs were the only cell types that were significantly responsive to both CS and reward on both days 1 and 7, PNs and SOM-INs also had significant responses to specific stimuli on certain days. While PNs did not show significant CS responses when compared to baseline, their reward responses on day 1 were significantly above the null distribution, and they became significantly lower than the null distribution on day 7 (*Figure 2E*). This result is in line with the change in tuning coefficient ($\rho_{reward}$), which showed a significant decrease in reward tuning between days 1 and 7 ($\bar{\Delta}\rho_{\text{reward}} = -0.141 \pm 0.067$; *Figure 6A*). Moreover, the cumulative distribution function of PN reliability also shifted significantly to lower reliabilities on day 7 compared to day 1 (*Figure 6B*). These results indicate that PNs initially responded to novel reward; however, they habituated to the reward following associative learning.

SOM-INs initially had no response to the CS on day 1, but their responses became significant on day 7 (*Figure 2H*). The change in CS tuning coefficient ($\rho_{tone}$) was not significant ($\rho_{tone} = .059 \pm 0.031$, *Figure 6C*), suggesting their responsivity did not change with learning. Interestingly, when we assessed the cumulative distribution of CS response reliability on days 1 and 7, the cumulative distribution function shifted to significantly higher reliability values on day 7 (*Figure 6C, D*). Notably, by day 7, there was a visible reduction in the number of SOM-INs that had 0% reliability to CS on day 1, indicating they were completely unresponsive to the CS on day 1 but not on day 7. Finally, SOM-INs showed modest but significant responses to reward on days 1 and 7 (*Figure 2H*). When we assessed the reward tuning among the SOM-IN population, $\rho_{reward}$ did not show a significant change between days 1 and 7 ($\rho_{reward} = . 0.017 \pm 0.040$, *Figure 6E*). However, SOM-IN reliabilities also shifted to higher values on day 7 (*Figure 6F*). Altogether, these results suggest that in naive mice, SOM-INs were unresponsive to the CS and modestly responsive to the novel reward; however, following associative learning, SOM-INs became more reliably responsive to both the CS and reward.

Lastly, reward consumption requires innate tongue movements during licking, and since microstimulation of M1 in mice has been shown to evoke tongue and jaw movements (*Komiyama et al., 2010*), it is crucial to distinguish whether the observed CS and reward responses resulted from task-related stimuli or if the activity is simply associated with licking movements. We demonstrated earlier that head-fixed mice learned the CS–reward association by displaying the conditioned response (anticipatory licking) following the CS on day 7 (*Figure 1*). To address this potential confound, we identified all the self-initiated licking bouts during ITIs, when no reward was present (*Figure 7A–C*; *Figure 2—figure supplement 1*). We first assessed all the significantly active cells in each cell type (identified in *Figure 2D*) during the first lick bout of each ITI on days 1 and 7. We observed that in each cell type, the majority of the neurons were nonlick neurons on both days 1 and 7 (*Figure 7—figure supplement 1A*). We then tracked individual lick and nonlick neurons and examined if they shifted their responses after associative learning. We found that most of the neurons maintained the same responses, and

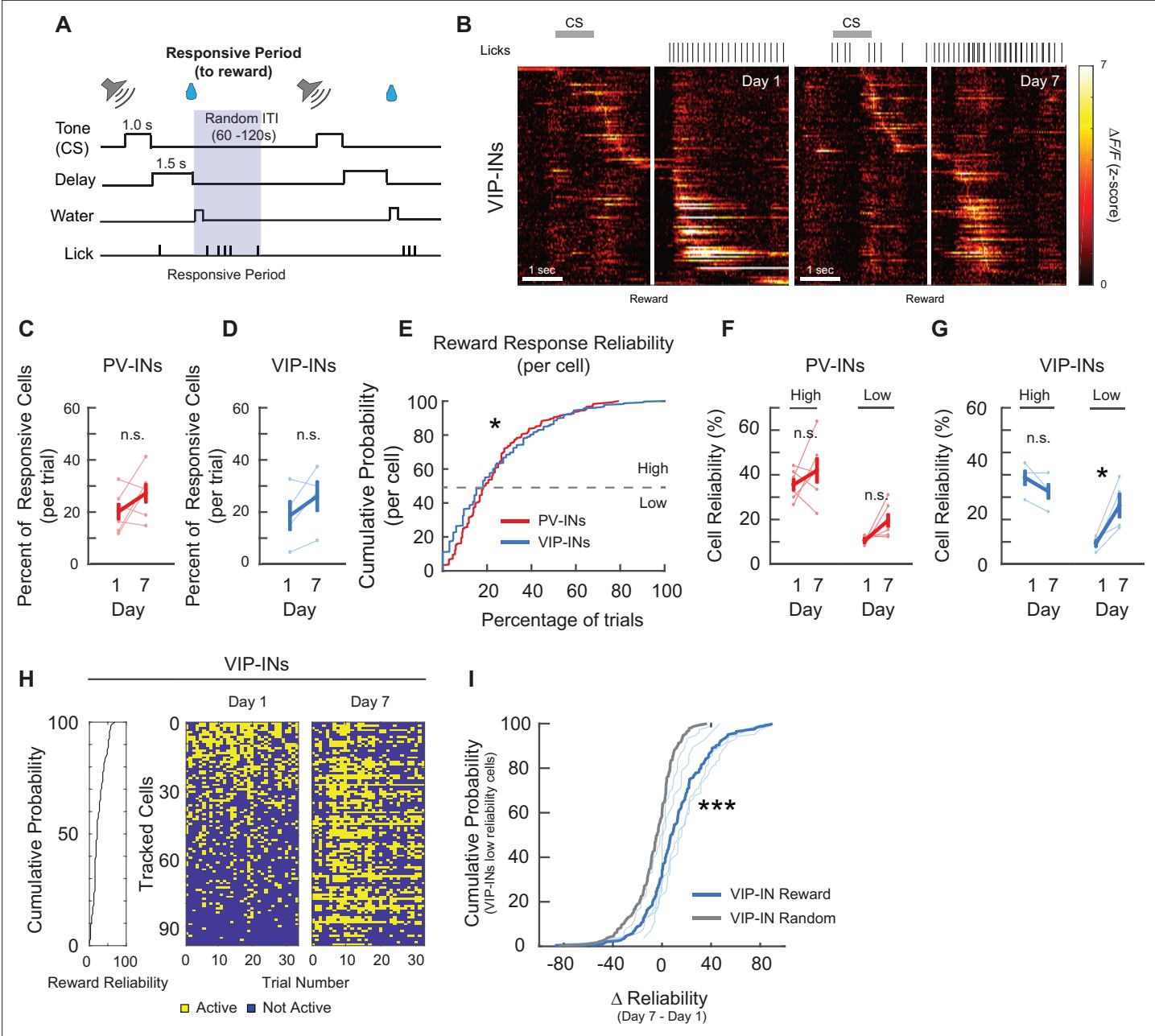

**Figure 5.** PV-IN and VIP-IN reward-related responses before and after associative learning. (**A**) Trial structure. Gray shaded bar represents the response period analyzed for reward-responsive activity. (**B**) Z-Scored activity of all the active VIP-INs from an example mouse during one representative trial on days 1 and 7, sorted by timing of maximum activity following the cue stimulus (CS) onset. Gray bar represents the timing of the CS. White line indicates the onset of water reward delivery. Mean percent of cells that are responsive to the reward for PV-INs (**C**) and VIP-INs (**D**). Neither PV- or VIP-INs showed a significant change. Paired *t*-test, n.s., nonsignificant, PV-IN: p = 0.16 (**C**), VIP-IN: p = 0.16 (**D**). (**E**) Cumulative probability plots showing the percent of trials that each neuron responded to reward for PV-INs and VIP-INs on day 1. Neurons from each cell type were pooled across mice. VIP-INs showed a significantly greater response reliability to the reward than PV-INs. Kolmogorov–Smirnov test, *p < 0.05, p = 5.2 × 10$^{-3}$. Mean reliability index of cells that are responsive to the reward for PV-INs (**F**) and VIP-INs (**G**). Each cell type is divided into High or Low Reliability Group based on the 50th percentile from the cumulative probability plots in (**E**). High and Low Reliability PV-INs maintained their consistency. (**G**) High Reliability VIP-INs did not show a change, while Low Reliability VIP-INs significantly increased in reliability following associative learning. Paired *t*-test, PV-IN High Reliability: p = 0.36, PV-IN Low Reliability: p = 0.058, VIP-IN High Reliability: p = 0.090, VIP-IN Low Reliability: p = 0.045. (**H**) Reward-responses from all tracked VIP-INs in an example mouse on days 1 and 7. Left, cumulative distribution of reward response reliability among all tracked cells within the example mouse. Right, binary map of each cell's reward response (active or not) across all trials on days 1 and 7. Cells were sorted by their day 1 reliability shown on the left and the order is maintained on day 7. Cells with low response reliability to reward on day 1 became more reliable on day 7. (**I**) Cumulative probability plots of the change in reliability from days 1 to 7 (*reliability$_{reward}$*) among VIP-INs with Low Reliability to reward on day 1. Bold blue, *reliability$_{reward}$* of the population.

*Figure 5 continued on next page*

*Figure 5 continued*

Thin blue lines show the $reliability_{reward}$ distribution within individual VIP-Cre mice. As a control, day 7 session was randomly sampled and a random reliability was calculated. $reliability_{random}$ was calculated by subtracting day 1 reward reliability from the day 7 random reliability (Gray, $reliability_{random}$ from the same population of VIP-INs). Kolmogorov–Smirnov test, ***p $< 1 \times 10^{-3}$ PV-IN: $n$ = 316 cells from six mice. VIP-IN: $n$ = 4 07 cells from four mice. Error bars show standard error of the mean (SEM).

nonlick neurons were still the majority in all cell types (*Figure 7—figure supplement 1B*). Next, we examined whether the lick neurons also showed mixed responses to CS, reward, or CS+ reward. Indeed, lick neurons exhibited mixed responses to CS, reward, or CS+ reward (*Figure 7—figure supplement 1C*). We then further divided them into three categories – 'CS cells', 'reward cells', and 'CS+ reward cells' and compared the percentage of neurons in each category between days 1 and 7; we did not observe a significant difference (*Figure 7—figure supplement 1D*). Lastly, at the population level, we examined the response reliability index of all the active neurons during all ITI lick bouts and compared them to the response reliability index for the CS and reward. On both day 1 (when there was minimal anticipatory licking during the CS) and day 7 (when mice showed anticipatory licking), all cell types exhibited lower reliability index values for the ITI lick bouts compared to the CS and reward, indicating that the increase in task-related responses following water rewards was specific to the reward stimulus, and not licking movements (*Figure 7D–J*). Together, these results suggest that the cell-type-specific modifications observed between days 1 and 7 were not caused by licking movements.

## Discussion

M1 is known to be involved in motor initiation, movement kinematics, and motor learning. Recent studies have demonstrated reward-related activity in M1 using in vivo electrophysiological recordings in nonhuman primates (*Marsh et al., 2015*; *Ramakrishnan et al., 2017*; *Ramkumar et al., 2016*) and transcranial magnetic stimulation in human subjects (*Thabit et al., 2011*). However, whether CS- and reward-associated signals are represented among different neuronal cell types within the microcircuit in M1 is still unclear. Using chronic two-photon Ca$^{2+}$ imaging, combined with transgenic mouse lines and viral strategies to target different neuronal cell types, we demonstrated that during a conditioning task, all major cell types in M1 responded to either the CS, the reward, or both. Most notably, each cell type underwent distinct modifications after association learning. By tracking the same population of neurons, we revealed that the CS-responding population increased among PV-INs and individual cells responded more reliably to the CS following associative learning. On the contrary, VIP-INs became more reliable in response to reward. When mice underwent control behavioral paradigms where tone was not paired with reward and no associative learning occurred, PV-INs and VIP-INs did not undergo these changes. Additionally, PNs had a drastically reduced response to reward, while SOM-INs became more reliable to both the CS and the reward. Our findings suggest that each cell type has a distinct role in processing information related to the cue–reward association in M1, and they may work together to provide the reinforcement signals in M1 that are important for motor skill learning.

Previous studies in trained rhesus monkeys performing a joystick center-out task have shown a widespread representation of reward anticipation and reward-related activity among cortical neurons in M1 (*Ramakrishnan et al., 2017*). Consistent with earlier work, we also observed reward-related activity in all four major cell types in M1, even in naive mice on day 1 when they were first exposed to the CS and reward. It has been reported that in sensory cortices, repeated passive exposure to a sensory stimulus leads to a long-lasting reduction in PN responsivity, but when animals are engaged in learning, PNs maintain their responsivity to the repeated stimulus (*Kato et al., 2015*; *Makino and Komiyama, 2015*). However, we found in M1, when water-restricted mice were engaged in a conditioning task to learn the association between the CS and water reward, PNs still showed a drastic habituation to the reward stimulus. A recent study that imaged neuronal activity in expert mice performing a head-fixed pellet reaching task demonstrated that L2/3 PNs in M1 are involved in encoding movement outcome (success vs. failure) but not the appetitive outcome (reward vs. no reward). However, the authors did not image the mice at the naive stage (*Levy et al., 2020*). Hence, one possibility is that L2/3 PNs in M1 encode reward signals during the naive stage, but after associative learning, they habituate and become unresponsive to the reward stimulus. In addition, in the sensory cortices, the

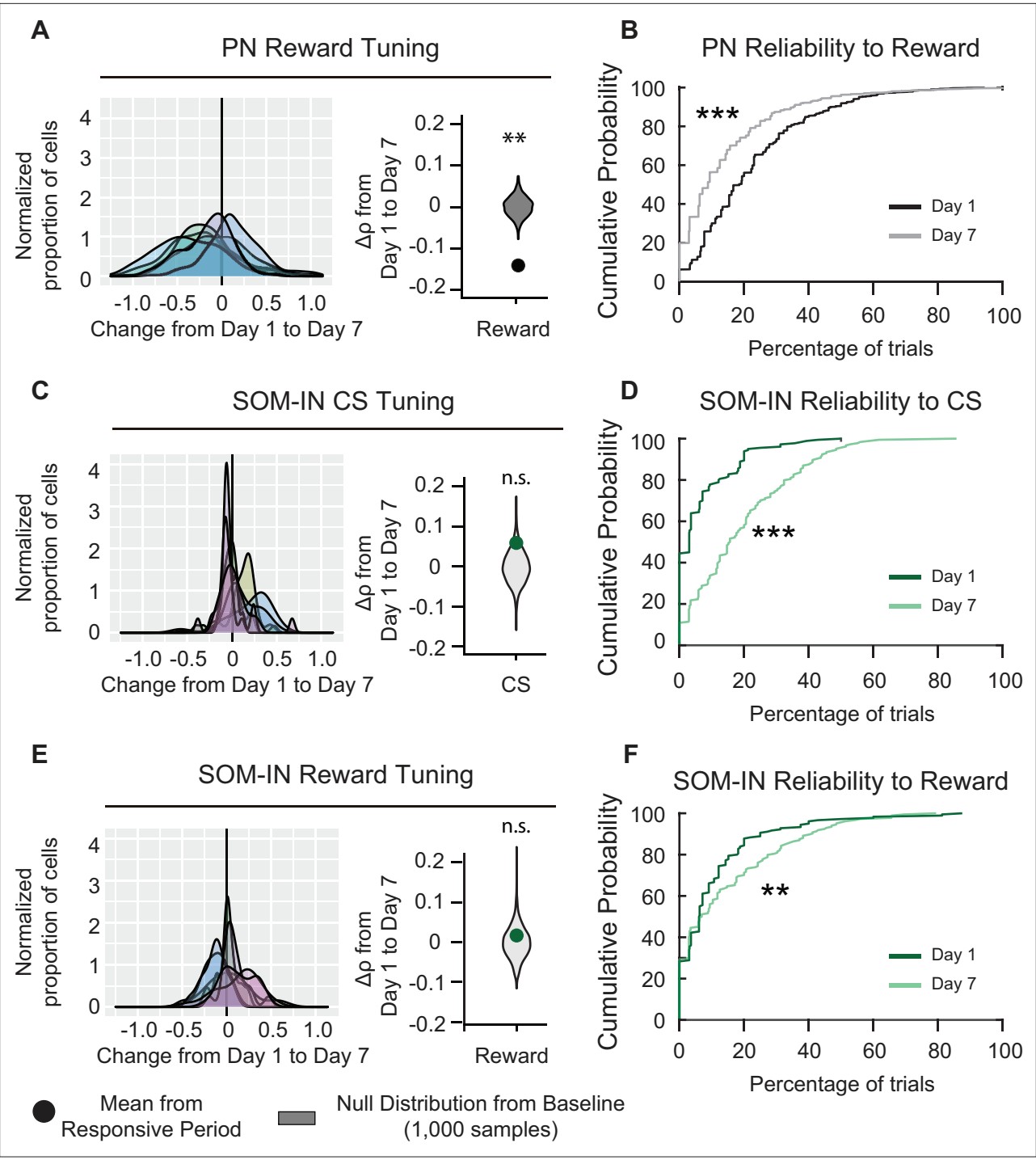

**Figure 6.** Pyramidal neuron (PN) and SOM-IN reliability is altered after associative learning. (**A**) Left, distribution of changes in PN Spearman correlation $\Delta\rho$ for reward. Each curve represents a Gaussian kernel density estimate of the distribution of $\Delta\rho$ in a single mouse. Right, mean change in Spearman correlation $\bar{\Delta}\rho$ . Null distributions (gray) were estimated by resampling each mouse and shuffling trials 1000 times. Reward tuning among PNs decreased after associative learning, Monte-Carlo, ***$p < 1 \times 10^{-3}$. (**B**) Cumulative probability plots showing the percent of trials that each neuron responded to reward for PNs on days 1 and 7. Neurons were pooled across mice. Day 7 reliability was significantly lower than day 1. Kolmogorov–Smirnov test, ***$p < 1 \times 10^{-3}$. (**C**) Left, distribution of changes in SOM-IN Spearman correlation $\Delta\rho$ with cue stimulus (CS). Each curve represents a Gaussian kernel density estimate of the distribution of $\Delta\rho$ in a single mouse. Right, mean change in Spearman correlation $\bar{\Delta}\rho$ . SOM-INs did not show a change in tone tuning. Monte-Carlo, n.s., nonsignificant, p = 0.128. (**D**) Cumulative probability plots showing the percent of trials that each neuron responded to the CS for SOM-INs on days 1 and 7. Neurons were pooled across mice. Day 7 reliability was significantly greater than day 1. Kolmogorov–Smirnov test, $p < 1 \times 10^{-3}$. (**E**) Left, distribution of changes in SOM-IN Spearman correlation $\Delta\rho$ with reward. Each curve represents a Gaussian kernel

*Figure 6 continued on next page*

*Figure 6 continued*

density estimate of the distribution of Δρ in a single mouse. Right, mean change in Spearman correlation $\overline{\Delta\rho}$ . SOM-INs did not show a change in reward tuning. Monte-Carlo, n.s., nonsignificant, p = 0.598. (**F**) Cumulative probability plots showing the percent of trials that each neuron responded to the reward for SOM-INs on days 1 and 7. Neurons were pooled across mice. Day 7 reliability was significantly greater than day 1. Kolmogorov–Smirnov test, **p = 0.012. PN: *n* = 1029 cells from six mice. SOM-INs: *n* = 189 cells from seven mice.

flexibility to either respond to or ignore sensory stimuli is based on the stimulus' behavioral relevance and is gated by local SOM-INs (*Kato et al., 2015*; *Makino and Komiyama, 2015*; *Poort et al., 2021*). In line with these findings, we found that SOM-INs became more reliably responsive to both the CS and the reward with associative learning. We also observed stimulus-specific increases in the reliability of PV-INs' response to the CS and VIP-INs' response to the reward after associative learning, and these changes to tone and reward were not observed in the absence of associative learning. When mice were exposed to tone alone, PV-INs did not show significant responses to tone on either day 1 or 7, while mice exposed to a nonpaired tone and reward task showed significant responses to tone on day 1, but not on day 7. This suggests that in M1, PV-INs only respond to behaviorally relevant cues such as those that predict reward. Finally, while VIP-INs remained responsive to reward in the nonpaired paradigm, VIP-INs did not show changes in reliability in the absence of associative learning. Together, our results suggest that different IN subtypes may have distinct roles in processing CS- and reward-related information in M1 during motivated associative learning.

One hypothesis is that PV-INs are recruited by the CS to control the behavioral responses (anticipatory licking) during reward anticipation since PV-INs are known to regulate PN firing through both feedforward and feedback inhibition (*Fishell and Rudy, 2011*; *Xu and Callaway, 2009*; *Xue et al., 2014*). Similar observations have been reported in the striatum, in which optogenetic activation or suppression of PV-INs during a similar conditioning task impaired anticipatory licking, demonstrating the importance of PV-INs in the expression of conditioned responses (*Lee et al., 2017*). Likewise, PV-INs in the basolateral amygdala, are also recruited during the CS and subsequently inhibited during the US in an auditory fear conditioning task. Optogenetic activation of PV-INs during the CS increased conditioned freezing behavior while PV-IN suppression reduced freezing, indicating bidirectional control of the conditioned response (*Wolff et al., 2014*). Our results demonstrate that in a naive animal, a subset of PV-INs in M1 are responsive to the CS only when rewards are present, and more PV-INs are recruited by the CS if the animal learns that the CS predicts reward. This suggests that in M1, PV-IN responses to the CS are not purely sensory, but rather, they may play an important role in controlling the behavioral responses to the CS.

VIP-INs, on the other hand, were significantly less reliable in responding to the CS compared to PV-INs, and their responses to the CS remained low. However, VIP-INs' responses to the reward were more reliable than those of PV-INs, and they became more closely tuned and reliably responsive to the reward with learning. Due to the disinhibitory position of VIP-INs in the microcircuit, activation of VIP-INs can lead to widespread increases in local excitability and contribute to regulating cortical gain (*Fu et al., 2014*; *Jackson et al., 2016*; *Pfeffer et al., 2013*). Furthermore, a growing body of evidence suggests a general principle across brain regions, in which VIP-INs receive long-range inputs (*Duan et al., 2020*; *Krabbe et al., 2019*; *Turi et al., 2019*; *Zhang et al., 2014*; *Gasselin et al., 2021*), respond to reinforcement signals (*Krabbe et al., 2019*; *Pi et al., 2013*), and play an important role in goal-oriented learning (*Krabbe et al., 2019*; *Turi et al., 2019*). Taken together, our results suggest that during CS–reward conditioning, PV-INs in M1 encode the CS association, and may regulate local circuit activity related to reward anticipation, whereas VIP-INs act as a context-dependent switch following the reward delivery (*Muñoz et al., 2017*; *Turi et al., 2019*) to instruct and disinhibit local PNs to enable learning-induced plastic changes critical for the acquisition of new movements. An interesting point to note is that since PV-INs are only responsive to tone when it is paired with reward, and neither PV-INs nor VIP-INs undergo plastic changes in the absence of associative learning, M1 is unlikely to be a primary site for learning reward predictions. We hypothesize that other brain regions are responsible for learning relevant CS–reward associations while filtering out behaviorally irrelevant stimuli, and these regions subsequently send long-range inputs to M1 to instruct motor responses to the CS and the reward. In summary, this study provides insight on how different IN subtypes in M1 integrate incoming inputs from various brain regions and orchestrate local circuit plasticity. Future

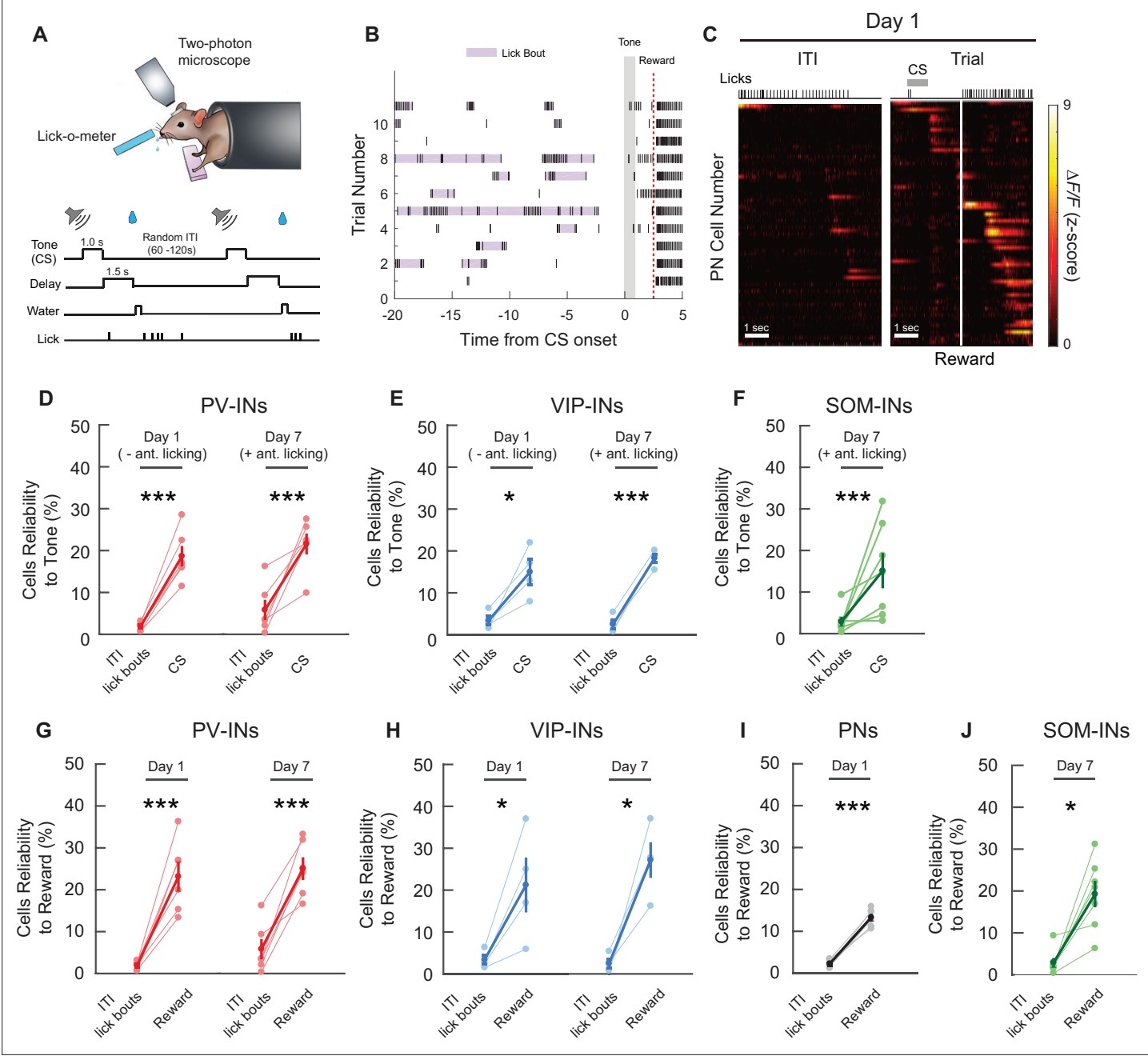

**Figure 7.** Cell-type-specific cue and reward activity are not due to licking movements. (**A**) Schematic of the cued reward conditioning task (top) and the trial structure (bottom). (**B**) Example licking behavior during the intertrial intervals (ITIs) from one mouse on day 1. Purple shading shows licks that were considered to be an individual lick bout. Gray shaded bar shows the cue stimulus (CS) timing and the red dotted line shows the reward timing. (**C**) Z-Scored activity of all the active pyramidal neurons (PNs) from an example mouse during one representative ITI lick bout. Left: maximum activity aligned to the lick bout onset. Right: maximum activity aligned to the CS onset. Gray bar represents the timing of the CS. White line indicates the time of water reward delivery. (**D–F**) Mean reliability index for all cell types with significant CS-related responses during ITI lick bouts with no water reward present, compared to the mean reliability index during the CS and up to but not including the reward delivery time. As shown in *Figure 1*, mice display anticipatory licking during the CS on day 7, but not day 1. PV-INs, VIP-INs, and SOM-INs were more reliably responsive during the CS than during licking movement alone. Only sessions/cell types with significant CS-related responses were analyzed. Paired *t*-test, *p < 0.05, ***p < 0.001. PV-INs day 1: p < $1 \times 10^{-3}$, PV-IN day 7: p = $6.6 \times 10^{-3}$ (**D**), VIP-IN day 1: p = 0.0355, VIP-IN day 7: p < $1 \times 10^{-3}$ (**E**), SOM-IN day 7: p = $3.9 \times 10^{-3}$ (**F**). (**G–I**) Mean reliability index for all cell types with significant reward-related responses during ITI lick bouts with no water reward present, compared to the mean reliability index following reward timing. All cell types were more reliably responsive during the reward period than during licking movement alone. Only sessions/cell types with significant reward-related responses were analyzed. Paired *t*-test, PV-INs day 1: p = $1.6 \times 10^{-3}$, PV-IN day 7: p = $1.6 \times 10^{-3}$ (**G**), VIP-IN day 1: p = 0.049, VIP-IN day 7: p = 0.014 (**H**), PN day 1 p < $1 \times 10^{-3}$ (**I**), SOM-IN day 7: p = 0.030 (**J**). PN: n = 1029 cells from six mice. PV-IN: *n* = 316 cells from

*Figure 7 continued on next page*

*Figure 7 continued*

six mice. VIP-IN: *n* = 407 cells from four mice. SOM-IN: *n* = 189 cells from seven mice. Error bars show standard error of the mean (SEM).

The online version of this article includes the following figure supplement(s) for figure 7:

**Figure supplement 1.** Lick cells were a small subset of active cells with stable mixed selectivity.

work will be important to identify the origin of these putative long-range inputs to different cell types in M1.

## Materials and methods

### Mice

Experimental mice were group housed in plastic cages with food and water ad libitum in a room with a reversed light cycle (12–12 hr). PV-Cre (008069), SOM-Cre (013044), VIP-Cre (010908), and B6129SF1/J (101043) mouse lines were acquired from Jackson Laboratory (Bar Harbor, ME, USA). All mouse lines were homozygous and in C57BL/6 × 129S4 background. For all mouse lines, both male and females were used. Mice were between P40 and P60 at the time of surgery.

### Surgery

Mice were deeply anesthetized under 1–2% isoflurane and given subcutaneous injections of Baytril (10 mg/kg) to prevent infection and buprenorphine (0.05 mg/kg) for analgesia. An incision was performed to remove a piece of the scalp and a custom head-plate was implanted onto the skull using instant glue (Krazy Glue) and dental cement (Lang Dental, Wheeling, IL, USA). A craniotomy of approximately 2 mm in diameter was performed over the right primary motor cortex. Virus (PNs: AAV1. CaMKII.GCaMP6f.WPRE.SV40; PV-IN, VIP-IN, and SOM-IN: AAV1.Syn.Flex.GCaMP6f.WPRE.5v40) was diluted 1:5 in saline and injected at a depth of ~250 µm from the pia using a glass pipette. All virus was obtained from Addgene (Watertown, MA, USA). Injections were performed at five sites, centered on coordinates 1.5 mm lateral and 0.3 mm anterior to bregma. For PN groups, 20 nl per site was injected. For PV-IN, VIP-IN, and SOM-IN, 40 nl per site was injected. All injections were performed at a rate of 10 nl/min and the pipette was left in place for 4 min following the injection to avoid back-flow. A glass imaging window was then implanted over the craniotomy and sealed with dental cement. Following surgery, a subcutaneous injection of dexamethasone (2 mg/kg) and buprenorphine (0.1 mg/kg) was given. Mice were given a minimum of 1 week to recover prior to beginning water restriction.

### Auditory cued reward conditioning behavior

Mice were gradually water restricted down to ~1 ml/day and were maintained at ~80% of original body weight over 2 weeks prior to the start of imaging/behavior sessions (*Chen et al., 2015*; *Harvey et al., 2012*; *Komiyama et al., 2010*; *O'Connor et al., 2013*; *Peters et al., 2014*). Mice were then head-fixed for simultaneous two-photon imaging and exposed to the conditioned stimulus (a constant 9 kHz auditory tone, 1 s in duration) followed by a 1.5-s delay period and a water reward (~10 µl). All lick times were measured by an infrared beam lick-o-meter and logged using the data acquisition software WaveSurfer (https://wavesurfer.janelia.org/). The ITI between the previous water reward and subsequent CS onset was randomly varied between 60 and 120 s. Each session was 1 hr in duration with 30–35 trials in total. Mice underwent 1 session/day for seven consecutive days. Two-photon calcium image was performed simultaneously on days 1 and 7 of the behavioral task.

To assess licking behavior, lick rate (number of licks per second, measured as infrared beam breaks) was calculated within 500-ms bins, then averaged across all trials within a session for each mouse. Lick rate was then averaged across mice. Mean anticipatory lick rate was calculated as the mean lick rate from the time of the CS onset to the end of the delay period (2.5 s in duration), not including the reward delivery. Mean ITI lick rate was calculated from the lick rate during the first 2.5 s of self-initiated spontaneous lick bouts. ITI lick bouts were defined as licking events that followed the previous trial by at least 20 s and preceded the subsequent trial by more than 2.5 s. Mean reward lick rate was calculated from the lick rate from the time of reward delivery to 2.5 s after.

All trials within a session were included in lick rate analysis in *Figure 1*. To ensure behavioral consistency across trials, only trials with at least three lick responses within 2.5 s of the reward delivery time were included in all analysis of neural responses.

In the control experiments with water rewards omitted, mice were head-fixed and exposed to a constant 9 kHz auditory tone, 1 s in duration, followed by a randomly varied ITI between 15 and 25 s. In the nonpaired control experiments, mice were exposed to a constant 9 kHz auditory tone, followed by a randomly varied delay period between 40 and 80 s before delivery of a water reward (*Krabbe et al., 2019*). Water rewards were then followed by a 15- to 25-s ITI. In both tasks, each session was 45 min in duration with an average of 30 trials. Mice underwent 1 session/day for seven consecutive days.

## Calcium imaging and analysis

In vivo imaging was performed using a commercial two-photon microscope (B-scope, Thorlabs, Newton, NJ, USA) and a ×16 water immersion objective (Nikon) with excitation at 925 nm (InSight X3, Spectra-Physics, Milpitas, CA, USA) with a frame rate of 30 Hz. Images were taken at 512 × 512 pixels covering 755 by 650 µm.

Images were corrected for movement in the x and y plane using full-frame cross-correlation image alignment (Turboreg *Thévenaz et al., 1998* plug-in ImageJ). The entire session was visually inspected and regions of interests (ROIs) were manually drawn on neurons using a custom MATLAB program, described in *Peters et al., 2014*. The ROI template from day 1 was loaded onto day 7 and aligned along the x and y plane. Only ROIs that could be tracked from days 1 to 7 were included in the dataset unless otherwise specified.

Fluorescence within an ROI was averaged across pixels. $\Delta F$ was calculated by subtracting a time-varying baseline fluorescence estimate ($F_0$) from the raw fluorescence trace. The calculation for baseline fluorescence ($F_0$) was calculated iteratively and based on inactive parts of the fluorescence trace as previously described (*Chu et al., 2016*; *Kato et al., 2012*; *Peters et al., 2014*; *Peters et al., 2017*).

We adapted a method by *Driscoll et al., 2017* to identify significant activity events for each neuron and then excluded ROIs with no significant activity events within the session, irrespective of the behavior. For each neuron the $\Delta F$ trace was circularly shifted by a random integer 1000 times and compared to the original trace. If the original $\Delta F$ trace was greater than the shifted data for at least five consecutive frames in at least 950 iterations, this was considered an active event. If a neuron did not have at least one active event in the entire session, irrespective of the behavior, it was removed from the dataset. This only accounted for a small proportion of ROIs as most of them are active on both days 1 and 7, as shown in *Figure 2D*.

For all subsequent analyses, a modified Z-score, adapted from *Kato et al., 2015*, was applied to $\Delta F$. The Z-score was calculated as $Z = (f(t) − \mu)/\sigma$, where $f(t)$ is the $\Delta F$ trace for a neuron, $\mu$ is the mean, and $\sigma$ is the standard deviation of the neuron's $\Delta F$ during the baseline period. The baseline period was a concatenation of 2.5 s preceding the CS onset (start of a trial) for all trials within a session.

## Calculation of tuning coefficients

We quantified the tuning of individual neurons to the CS and reward stimuli delivered in our classical conditioning task using the nonparametric Spearman correlation $\rho$ (scipy.stats.spearmanr) between the trial-averaged fluorescence and the timing of stimulus delivery

$$\rho_{m,n,s}^{(d)} = \text{corr}\left[\frac{1}{|\mathfrak{T}_m^{(d)}|}\sum_{t \in \mathfrak{T}_m^{(d)}} \mathbf{f}_{m,n,t-T_{\text{baseline}} \,:\, t+T_{\text{CS}}+T_{\text{delay}}+T_{\text{reward}}+T_{\text{post}}}^{(d)}, 1_s\right],$$

where $t$ is the start time of a trial (defined as the start of the CS), $\mathfrak{T}_m^{(d)}$ is the set of all trial start times from mouse $m$ on day $d \in \{1, 7\}$, $\mathbf{f}_{m,n,t-T_{\text{baseline}} \,:\, t+T_{\text{CS}}+\cdots}^{(d)}$ is the fluorescence trace of neuron $n$ from mouse $m$ during a single trial, $1_s$ is an indicator function for stimulus $s \in \{\text{CS}, \text{reward}\}$, $|\mathfrak{T}_m^{(d)}|$ is the number of trials, and $\rho$ is the Spearman correlation coefficient. Analysis was carried out with $T_{\text{baseline}} = 2$ s and $T_{\text{post}} = 6$ s. We considered the 'CS' period indicated by $1_{\text{CS}}$ to range from the start of the CS at time $t$ to the start of reward delivery at time $t + T_{\text{CS}} + T_{\text{delay}}$, and the 'reward' period indicated by $1_{\text{reward}}$ to

be the first 2.5 s of reward delivery (see schematic in **Figures 4A and 5A**). We used the change in $\rho$ from days 1 to 7 as a cell-resolved measure of changes in tuning over the course of learning.

To summarize learning-associated changes in tuning, we calculated the mean change in the Spearman correlation for each cell type and trial component (CS or reward) from days 1 to 7 as follows

$$\bar{\Delta}\rho_s = \tfrac{1}{|\mathfrak{M}|} \sum_{m \in \mathfrak{M}} \tfrac{1}{N_m} \sum_{n=1}^{N_m} \left( \rho_{m,n,s}^{(7)} - \rho_{m,n,s}^{(1)} \right),$$

where $\mathfrak{M}$ is the set of mice used in the experiment, $N_m$ is the number of neurons in mouse $m$, and $\rho_{m,n,s}^{(d)}$ is the Spearman correlation as defined above.

We used a nonparametric approach for statistical tests involving the mean change in Spearman correlation by scrambling trial times and bootstrapping mice to construct a null distribution for $\bar{\Delta}\rho_s$ . Specifically, we first drew a random sample of $|\mathfrak{M}|$ mice from $\mathfrak{M}$ with replacement, then drew a random sample of $|\mathfrak{T}_m^{(d)}|$ trial start times uniformly distributed between 0 and $T_{\text{session}}^{(d)} - (T_{\text{baseline}} + T_{\text{CS}} + T_{\text{delay}} + T_{\text{reward}} + T_{\text{post}})$ for each day $d$ and randomly selected mouse, and finally used these randomly selected mice and scrambled trial start times to compute the change in tuning $\bar{\Delta}\rho_s$ . This process was repeated 1000 times to approximate the distribution of $\bar{\Delta}\rho_s$ under the null hypothesis that changes in tuning are unrelated to the CS and reward delivery. We considered the observed changes in tuning $\bar{\Delta}\rho_s$ to be statistically significant at the $*$ or $**$ level if they fell into the 5 or 1% tails of this distribution, respectively.

## Activity analysis

To identify neuron responses to the CS and reward, we applied a set threshold to each neuron. Neurons were defined as CS or reward responsive on a trial-by-trial basis if they exceeded 1 $Z$-score (excitation threshold used in **Kato et al., 2015**) for at least five consecutive frames within 2.5 s of the CS onset or 2.5 s of the reward delivery time, respectively. This was assessed for each trial with at least three lick responses within 2.5 s of the reward delivery time. We then took the median of the percent of responsive neurons across all trials in a session from one mouse, and the mean across mice. In the nonpaired experiments, since the reward was not preceded by a CS, the mice required variable amounts of time to notice the water reward. Therefore, in the nonpaired experiments, we calculated reward responses within 2.5 s from the onset of the first lick following the water delivery. In the case that no licks were recorded before the subsequent tone, the trial was not included in the reward analysis.

We used a Monte-Carlo approach to validate the percent of CS- and reward-responsive neurons. The mean percentage of CS- and reward-responsive neurons observed were compared to a null distribution made for each cell type on each day. We randomly sampled mice with replacement, then sampled the entire session, and then calculated the percentage of active cells (exceeding 1 $Z$-score for at least five consecutive frames) during a randomly chosen 2.5-s window. For each mouse, the number of samples was equal to the number of included trials (i.e., number of trials with at least three lick responses within 2.5 s of reward delivery). We then took the median across the random samples and then took the mean across mice to obtain a mean percentage of responsive neurons during a randomly chosen time window. This was repeated 1000 times to generate a null distribution of mean percentage of active neurons. To assess whether the observed percentage of CS- and reward-responsive neurons was significantly different from the null distribution, the observed value was compared to the tails of the null distribution. This was done for each cell type on both days 1 and 7. We considered the CS or reward responses to be statistically significant at the $*$ or $**$ level if they fell into the 5 or 1% tails of this distribution, respectively, and *** if there was no overlap with the distribution. Since this approach tests the null hypothesis that the observed neuronal responses are due to chance (in this case, baseline activity/noise), only cell types with a significantly higher percentage of responsive neurons for a given session were analyzed further.

The CS/reward reliability index was defined as the percentage of trials within a session where the neuron was CS/reward responsive. The reliability cumulative distribution was made by pooling the day 1 index values of all the neurons from a neuronal cell type (across mice). If a neuron's day 1 index value was lower or equal to the index value at the 50th percentile of the cumulative distribution (excluding nonresponsive neurons with a reliability of 0) for that cell type, it was categorized into the Low Reliability group. If a neuron's day 1 index value

exceeded the 50th percentile value, it was categorized into the High Reliability group. To assess changes in reliability at the population level, we took the mean reliability within each group on days 1 and 7. To assess changes in reliability among individual Low Reliability neurons, we used a *reliability* measure where, $reliability_{CS} = (CS\ reliability_{Day\ 7} - CS\ reliability_{Day\ 1})$ and $reliability_{reward} = (reward\ reliability_{Day\ 7} - reward\ reliability_{Day\ 1})$. As a control, we randomly sampled the day 7 session matching the number of trials, and calculated the reliability to obtain a 'random reliability' for each neuron. We then calculated a $reliability_{random}$ where, $reliability_{Random} = (reliability_{random} - reliability_{Day\ 1})$ for CS and reward reliabilities.

The onset time of neuronal activity following the CS was calculated as the time from the CS onset to the time of the first $Ca^{2+}$ event (fifth frame above threshold) within the CS response period. For the onset time of reward-related neuronal activity, the time from the first lick (after reward delivery) to the time of the first $Ca^{2+}$ event (fifth frame above threshold) within the reward response period was used. The latency for each cell was first calculated by taking the mean across all active trials for a single cell, then the median of all cells within a mouse was calculated. Only cells that were tracked between days were included.

To determine the proportion of PV-INs that responded to the CS on days 1 and 7, we found the overall proportion of cells that responded to the CS out of total active cells on both days 1 and 7. To calculate the percentage of PV-INs that maintained CS responses across days, we found the proportion of day 7 cells that also had CS responses on day 1. We also found the proportion of day 1 cells with CS responses, that either maintained CS responses on day 7, became CS unresponsive but reward responsive, or became unresponsive to both CS and reward. The same analysis was performed on VIP-IN reward responses. Only active cells that were tracked from days 1 to 7 were included.

## Licking-related analysis

ITI lick bouts were defined as self-initiated licking events that occurred at least 20 s after the preceding reward delivery time (trial end) and more than 2.5 s prior to the subsequent CS onset (trial start). If individual licks were separated by 3 s or more, they were considered to be a new lick bout. To remain consistent with CS and reward analyses, only the first 2.5 s of a lick bout were analyzed for neural responses. ITI lick bout reliability indices were calculated as described above.

To determine lick cells, we found the first ITI lick bout in each ITI and calculated the mean $Z$-scored $\Delta F$ during the first 2.5 s of the lick bout. We then created a matrix of concatenated ITIs from the same session and randomly sampled the concatenated ITIs, and calculated the mean $Z$-scored $\Delta F$, matching the duration and number of ITI lick bouts. A paired $t$-test was used to compare the mean $Z$-scored $\Delta F$ during ITI lick bouts and during the random samples. Cells with significantly higher $\Delta F$ during lick bouts were considered to be lick cells. We performed this analysis on days 1 and 7 and tracked individual neurons to identify changes in selectivity on a cell-by-cell basis. To determine if cells have mixed selectivity, we performed the same analysis using the CS and reward response periods (2.5 s from onset) and compared this activity to an equal number of random samples using a paired $t$-test.

## Statistical analysis

Statistical analysis for tuning coefficients was performed in Python and in R. All other statistical analyses were performed in Matlab using the Statistics and Machine Learning Toolbox. Two-way analysis of variance (ANOVA) was used to test for differences in anticipatory lick rate on days 1 and 7. One-way ANOVA was used to test for differences in lick rate during ITI, CS, and reward. One-way ANOVA was used to compare the percent of active cells across cell types on a single day. Monte-Carlo (as described above) was used to test for significant percent of CS- and reward-responsive neurons, and for changes in tuning properties. Paired $t$-test was used to test for differences in the percentage of responsive cells and reliability index on days 1 and 7, and for differences in neuron reliability between ITI lick bouts, CS, and reward. Paired $t$-test was used to determine mixed selectivity as described above. The Kolmogorov–Smirnov test was used to compare response reliability cumulative distributions and Δreliability distributions. All values were reported as the mean and standard error of the mean unless otherwise specified. Power analysis was not performed to predetermine the sample size, and the experiments were not blinded.

## Data analysis and code availability

Tuning coefficient calculation and statistical tests were performed using Python 3.8 with the following libraries: NumPy, Pandas, h5py, and SQLAlchemy. Figures were prepared in Python using matplotlib and seaborn, and in R using ggplot2. Codes to reproduce the analysis for *Figures 1, 2, and 4–7* are available at https://githubcom/clee162/Analysis-of-Cell-type-Specific-Responses-to-Associative-Learning-in-M1. Codes to reproduce the analysis and *Figure 3* are available at https://githubcom/nauralcodinglab/interneuron-reward. Data can be found on Dryad at https://doiorg/105061/dryadq573n5tjj.

## Acknowledgements

We thank the members of the Chen lab for discussions and providing feedback on the manuscript. This work was supported by grants for S.X.C. from Canada Research Chair (CRC) (grant no. 950-231274) and Natural Sciences and Engineering Research Council of Canada (NSERC) (grant no. 05308), and a grant for R N from NSERC (grant no. 06972). E.H. was supported by a NSERC graduate scholarship. C.L. was supported by Ontario Graduate Scholarship and Queen Elizabeth II Graduate Scholarship.

## Additional information

### Funding

| Funder | Grant reference number | Author |
|---|---|---|
| Natural Sciences and Engineering Research Council of Canada | 05308 | Simon Chen |
| Canada Research Chairs | 950-231274 | Simon Chen |
| Natural Sciences and Engineering Research Council of Canada | 06972 | Richard Naud |

The funders had no role in study design, data collection, and interpretation, or the decision to submit the work for publication.

### Author contributions

Candice Lee, Conceptualization, Data curation, Formal analysis, Investigation, Methodology, Supervision, Validation, Visualization, Writing - original draft, Writing - review and editing; Emerson F Harkin, Formal analysis, performed the analyses for Figure 3, performed the analyses for Figure 3; Xuming Yin, Data curation; Richard Naud, Formal analysis, Supervision, performed the analyses for Figure 3, performed the analyses for Figure 3; Simon Chen, Conceptualization, Funding acquisition, Supervision, Writing - original draft, Writing - review and editing

### Author ORCIDs

Candice Lee ⓘ http://orcid.org/0000-0001-7043-9367
Emerson F Harkin ⓘ http://orcid.org/0000-0003-0698-5894
Richard Naud ⓘ http://orcid.org/0000-0001-7383-3095
Simon Chen ⓘ http://orcid.org/0000-0002-7595-7443

### Ethics

All animal experiments were approved by the University of Ottawa Animal Care Committee (protocol #: CMM-2737) and in accordance with the Canadian Council on Animal Care guidelines.

### Decision letter and Author response

Decision letter https://doi.org/10.7554/eLife.72549.sa1
Author response https://doi.org/10.7554/eLife.72549.sa2

## Additional files

### Supplementary files

• Transparent reporting form

• Supplementary file 1. Summary.of the number of mice, total regions of interests (ROIs), total active cells, and total active cells tracked from days 1 to 7 in all experimental conditions.

### Data availability

Codes to reproduce the analysis for figures 1-2 and 4-7 are available at https://github.com/clee162/Analysis-of-Cell-type-Specific-Responses-to-Associative-Learning-in-M1. Codes to reproduce the analysis and figure 3 are available at https://github.com/nauralcodinglab/interneuron-reward. Data can be found on Dryad at https://doi.org/10.5061/dryad.q573n5tjj.

The following datasets were generated:

| Author(s) | Year | Dataset title | Dataset URL | Database and Identifier |
|---|---|---|---|---|
| Lee C | 2021 | Cell-type specific responses to associative learning in the primary motor cortex | https://github.com/clee162/Analysis-of-Cell-type-Specific-Responses-to-Associative-Learning-in-M1 | Github |
| Chen SX | 2022 | Data from: Cell-type specific responses to associative learning in the primary motor cortex | http://dx.doi.org/10.5061/dryad.q573n5tjj | Dryad Digital Repository, 10.5061/dryad.q573n5tjj |
| Lee C | 2021 | Data from: Cell-type specific responses to associative learning in the primary motor cortexon-reward | https://github.com/nauralcodinglab/interneuron-reward | Github |

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
