## [Editor Report]

Using advanced live brain imaging techniques, the authors studied the activities of neurons in the primary motor cortex of mice during a classical conditional task, in which a tone is paired with a water reward. They found that distinct types of neurons respond differently to the auditory cue or the reward, and the responses evolve differentially as learning proceeds. This work reveals an interesting role of the motor cortex beyond its well-recognized function in motor control and suggests distinct functions of pyramidal neurons as well as various interneurons in reinforcement learning.

---

## [Decision Letter]

**Decision letter after peer review:**

Thank you for submitting your article "Cell-Type Specific Responses to Associative Learning in the Primary Motor Cortex" for consideration by *eLife*. Your article has been reviewed by 3 peer reviewers, and the evaluation has been overseen by a Reviewing Editor and Michael Frank as the Senior Editor. The following individuals involved in review of your submission have agreed to reveal their identity: Jerry L Chen (Reviewer #1); Hyung-Bae Kwon (Reviewer #3).

Essential revisions:

Reviewers think that additional control experiments are needed for the revision. In addition, additional data analysis and clarification are needed to strengthen the paper. The following are essential revisions:

1. To address the question of whether changes are specifically associated with learning, a critical control experiment is missing. The behavioral study lacks a non-paired control – it would be more compelling if there was a CS of the same modality that was not paired with the US so we can be sure that the effects are not cue-specific but specific to conditioning. Then those cues would need to be counterbalanced across animals. This would be important for us to conclude the neural effects reflect associative learning vs some other impact of the cue over time.

2. Please examine the motor-related activity in the data set and address the following questions raised by Reviewer #1 and Reviewer #2, (a) As a control, can you quantify the number of licking-related neurons across cell types and confirm that they do not change with learning? (b) Are there neurons that show mixed responses to cue and licking? Do those responses change at all during learning? (c) Are there neurons show mixed responses to licking and reward? Do those responses change at all during learning?

3. The 2.5s period chosen for analysis covers both tone presentation and delay. Will the conclusions in Figure 2E-H change if the analysis is restricted to the 1s of tone presentation? In addition, it seems better to use the actual duration of reward presentation (see Q2) for analysis. As the statistical analysis is done by random shuffling, there seems no need to match the period for tone and reward analyses.

4. In Activity Analysis and Tuning Coefficients Calculation, the authors performed a resampling of mice with replacement, and the size of the random sample equals that of the original set of mice. Please clarify why this is done this way. Can the authors simply take all mice into analysis?

5. Both Reviewer #2 and Reviewer #3 pointed out the importance of plotting calcium signals over days without resorting. The authors conclude that "PV-INs that began as highly reliable maintained their reliability to the CS, …, while PV-INs that began as low reliability became significantly more reliable." However, Figure 4F only shows how the percentage of neurons in the high- or low-reliability category changes overtraining. To draw this conclusion, the authors need to track individual neurons and compare the same neuron's reliability on d1 and d7.

*Reviewer #1 (Recommendations for the authors):*

I would encourage the authors to examine the motor-related activity in their data set, to help shed light on the following questions. Do the cue and reward related changes really reflect local circuit changes as the authors seem to suggest? Or could they potentially reflect changes outside of M1 that are then inherited and readout by M1 cell type? If local circuit changes are occur, one might expect to see changes in the conjunctive responses of cue, reward, and motor activity within individual cells. Changes in network activity between cue, reward, and motor cells may also be observed. It should be possible and worth examining these relationships to tease apart potential mechanisms and impact of non-motor changes in M1.

*Reviewer #2 (Recommendations for the authors):*

1. It is helpful to provide further details about the genetic background of the transgenic animals. Are they homozygous or heterozygous? What genetic background are they maintained in? Also, it would be helpful to indicate the number of neurons imaged per mouse.

2. A question about the behavioral setup. How long is the water reward presented? How many licks does it take the mouse to consume the 10 ul water reward? It seems from Figure 1B that most licks occur when water is no longer available. Also, the example in Figure 2C suggests that post-cue licking is far fewer on d7 than on d1, suggesting that the mouse has learned to inhibit non-rewarded licking. This seems not to agree with Figure 1C.

3. As imaging is conducted in M1, how is the response of the cells related to licking? Can the authors make a licking-triggered average of Ca traces for comparison?

4. In Figure 2B: is the scale bar 10% or 10, i.e., 1000%? Many transients show a very slow onset, which is not consistent with the rapid rising phase of GCaMP6f signals as shown in many previous publications. Also, previous publications show that PV and SOM interneurons have very synchronized activity in mPFC (Pinto and Dan, 2015 Neuron) and secondary motor cortex (Garcia-Junco-Clemente et al., 2019 Cell Report). Is it true in M1? Can the authors give some examples of Ca traces of each type of the interneurons? If Ca transients in M1 interneurons exhibit different kinetics from those in pyramidal neurons, how would that affect the choice of analysis criteria?

5. The 2.5s period chosen for analysis covers both tone presentation and delay. Will the conclusions in Figure 2E-H change if the analysis is restricted to the 1s of tone presentation? In addition, it seems better to use the actual duration of reward presentation (see Q2) for analysis. As the statistical analysis is done by random shuffling, there seems no need to match the period for tone and reward analyses.

6. In Activity Analysis and Tuning Coefficients Calculation, the authors performed a resampling of mice with replacement, and the size of the random sample equals that of the original set of mice. What is the reason to do this? Can the authors simply take all mice into analysis?

7. Figure 3A-B: a little clarification is needed. (1) Is each trace a single trial or the average over trials? (2) What does it mean that "each trace is from the same neuron on d7?" Are the five traces from one neuron, or five neurons? If they are from five neurons, are these the same five neurons in A and B? (3) Why does the color of a single trace change with time?

8. The authors conclude that "PV-INs that began as highly reliable maintained their reliability to the CS, …, while PV-INs that began as low reliability became significantly more reliable." However, Figure 4F only shows how the percentage of neurons in the high- or low-reliability category changes over training. The authors need to track individual neurons and compare the same neuron's reliability on d1 and d7 to draw this conclusion. The same issue applies to the analysis of all cell types.

*Reviewer #3 (Recommendations for the authors):*

First of all, the work is great. Analyzing calcium activity from each interneuron cell type is quite difficult, but authors elegantly performed the experiments. I wonder whether you can plot calcium signals over days (day 1 to day 7) without resorting. I understand it is very difficult, but if you have a good imaging quality enough to trace calcium activity from the same cells over several days, it would be nicer to show. Another concern is the lack of control experiments that show no such changes presented in training groups. Otherwise, it is quite good.

---

## [Author Response]

Essential revisions:Reviewers think that additional control experiments are needed for the revision. In addition, additional data analysis and clarification are needed to strengthen the paper. The following are essential revisions:1. To address the question of whether changes are specifically associated with learning, a critical control experiment is missing. The behavioral study lacks a non-paired control – it would be more compelling if there was a CS of the same modality that was not paired with the US so we can be sure that the effects are not cue-specific but specific to conditioning. Then those cues would need to be counterbalanced across animals. This would be important for us to conclude the neural effects reflect associative learning vs some other impact of the cue over time.

We thank the reviewers for suggesting this control experiment. To properly address this, we conducted 3 sets of experiments using PV-Cre and VIP-Cre mice. In the first experiment, we exposed PV-Cre mice to the same auditory tone used as the CS but we omitted all water rewards, and recorded the response of PV-INs to tone on both Day 1 and Day 7 (Figure 4 —figure supplement 1A). Additionally, in separate cohorts of animals, we exposed PV-Cre and VIP-Cre mice to tone but with non-paired water rewards (given at randomly varied time intervals) and recorded the responses from PV-INs and VIP-INs on both Day 1 and Day 7 (Figure 4 —figure supplement 1D). In all three experiments, tone was presented at the same frequency and duration as previous CS-reward experiments, and the number of trials was also equivalent.

Surprisingly, we found that when mice were not given water rewards, PV-INs did not respond to the tone stimulus on either Day 1 or Day 7, as the mean percent of active cells during the tone response period was not significantly different from the null distribution generated by randomly sampling the session (Figure 4 —figure supplement 1B). In comparison, when we examined mice that were exposed to the tone stimulus with non-paired water rewards, we found that PVINs were significantly responsive to the tone stimulus on Day 1, similar to what we observed in Figure 2. Interestingly, by Day 7, PV-INs no longer responded to the tone stimulus (Figure 4 —figure supplement 1G). We next examined if the mice that received the tone stimulus with nonpaired water reward learned to associate the tone with the reward after 7 days. We found the animals did not learn the association, as their conditioned response (anticipatory licking) did not increase at Day 7 (Figure 4 —figure supplement 1E). Together, in the first set of experiments where reward was omitted, we demonstrated that PV-INs did not respond to the tone stimulus on either day in the absence of reward. In the second set of experiments, where the tone was not paired with reward and the mice did not learn the CS-reward association, PV-INs were initially responsive to the tone on day 1, but by day 7, they no longer responded. These experiments suggest that PV-INs in M1 do not respond to auditory tone in general, but instead only respond to the tone when the animal is actively associating it with reward. In addition, unlike the mice that learned the association, we did not observe a change in the mean percent of tone-responsive PV-INs from Day 1 to Day 7 in the non-paired mice, and ‘Low Reliability’ PVINs did not show a change in tone reliability (Figure 4 —figure supplement 1H-I). These results further demonstrate that the changes among PV-INs to the CS tone were specific to associative learning.

Lastly, in Figure 5, we showed that after associative learning, the reward response reliability of ‘Low Reliability’ VIP-INs increased; therefore, we performed an additional control experiment examining VIP-IN responses to reward in the non-paired paradigm. We found that although VIPcre mice did not learn to associate the tone with the random water rewards (Figure 4 —figure supplement 1E), VIP-INs consistently responded to rewards (Figure 4 —figure supplement 1M). However, without learning the association, the reward response reliability of ‘Low Reliability’ VIP-INs did not change from Day 1 to Day 7 (Figure 4 —figure supplement 1O).

In conclusion, these control experiments further support our findings that different cell types in M1 undergo cell-type specific modifications after associative learning, in which PV-INs’ responses became more reliable to the cue stimulus, while VIP-INs’ responses became more reliable to the reward. We have included these results in the manuscript (Line 235 – 260, 279 – 290).

2. Please examine the motor-related activity in the data set and address the following questions raised by Reviewer #1 and Reviewer #2, (a) As a control, can you quantify the number of licking-related neurons across cell types and confirm that they do not change with learning? (b) Are there neurons that show mixed responses to cue and licking? Do those responses change at all during learning? (c) Are there neurons show mixed responses to licking and reward? Do those responses change at all during learning?

We thank the reviewers for these suggestions. To identify licking-related neurons, we took all the active cells within a session and examined their activity during the first lick bout of each ITI. We then took the mean z-score during the lick bouts and compared it to the mean z-score of an equal number of randomly sampled bouts (with equal duration). If the mean lick bout z-score was significantly higher than the mean random bout z-score, we considered the cell to be a lick neuron. We found that on both Day 1 and Day 7, the majority of neurons (from all cell types) were non-lick neurons (Figure 7 —figure supplement 1A). We also tracked individual neurons and performed the same analysis for both Day 1 and Day 7 and examined whether lick and nonlick neurons shift their responses after associative learning. We found that the majority of cells maintained the same responses, and non-lick cells were still the largest group in all cell types (Figure 7 —figure supplement 1B).

Next, we examined whether these lick neurons also showed mixed responses to CS and reward on Day 1 and Day 7. Indeed, these lick neurons exhibited mixed responses to CS, reward, or CS + reward (Figure 7 —figure supplement 1C). We then divided the lick neurons into three categories – ‘CS cells’, ‘reward cells’, and ‘CS + reward cells’. When we compared the percentage of neurons in each category between Day 1 and Day 7, we did not observe a significant difference (Figure 7 —figure supplement 1D). Together, our analyses showed that there were not many lick neurons (from all cell types) present in M1, and the percentage of lick and non-lick neurons did not change after associative learning. This is consistent with previous optical and electrical micro-stimulation studies that showed tongue, jaw, and lip are more reliably evoked in the Anterior-Lateral-Motor area (ALM) (PMID: 20376005, 2036613), rather than M1. Furthermore, we found that while these lick neurons also exhibit mixed responses to CS, reward, or CS + reward, the percentage of these mixed response cells did not change after associative learning. Therefore, since the lick cells were stable from Day 1 to Day 7, it is unlikely that they impacted the learning related changes. We have included these results in Figure 7 —figure supplement 1 and Line 322 – 331.

3. The 2.5s period chosen for analysis covers both tone presentation and delay. Will the conclusions in Figure 2E-H change if the analysis is restricted to the 1s of tone presentation? In addition, it seems better to use the actual duration of reward presentation (see Q2) for analysis. As the statistical analysis is done by random shuffling, there seems no need to match the period for tone and reward analyses.

Following the reviewers’ suggestion, we calculated the percent of CS-responsive neurons for each cell type on Day 1 and Day 7 using a 1s time window following CS presentation. Surprisingly, we found the percentage of CS-responsive neurons decreased significantly compared to the 2.5s window in all cell types, and the percentages were all around 5% of total active neurons (Author response image 1). One plausible explanation is that M1 is not a region for direct sensory input related to auditory tone, water, or reward; hence, neuronal responses in M1 could be slightly delayed. This is supported by our no-reward control experiment, in which PV-INs did not respond to auditory tone when no water rewards were given (Figure 4 —figure supplement 1B). In combination with the slow kinetics of calcium indicator compared to electrophysiology recording, we believe this explains why a 1s window is not sufficient to capture CS or reward-related activity in M1. Since all cell types only have around 5% of CS-responsive cells, we do not think using the 1-sec time window truly represents the CS responses in M1.

**Author response image 1. sa2fig1:** (A) Comparing the percent of CS-responsive cells calculated with a 2. 5s or a 1.0s response window after CS onset. For all cell types, the percent of CS-responsive cells was greatly reduced when using a 1.0s response window for analysis compared to a 2.5s window. Most cell-types were reduced to about 5% of responsive cells. Paired t-test, * *p < 0.05*, ** *p <0.01*, *** *p < 0.001*, PN: Day 1 *p < 1*×*10^-3^*, Day 7 *p < 1*×*10^-3^*; PV-IN: Day 1 *p = 0.002*, Day 7 *p = 0.001*; VIP-IN: Day 1 p = 0.008, Day 7 *p < 1*×*10^-3^,* SOM-IN: Day 1 *p = 0.034,* Day 7 *p < 1*×*10^-3^* (B) Comparing Monte-Carlo resampling methods using sampling with replacement vs. without replacement. For each cell type, the null distribution estimates the percent of cells active at baseline using sampling with replacement (top row) and without replacement (bottom row) for Day 1 and Day 7. With replacement: mice were re-sampled with replacement, the session was randomly sampled, and the mean percent of active cells was calculated; this was repeated 1000 times. Without replacement: all mice were sampled once, the session was randomly sampled, and the mean percent of active cells was calculated; this was repeated 1000 times. As random sampling of mice with replacement allows for a more unbiased estimate of between animal variability, the distribution created without replacement is much narrower with a smaller range and standard deviation. PN: n = 1029 cells from 6 mice. PV-IN: n = 316 cells from 6 mice. VIP-IN: n = 407 cells from 4 mice. SOM-IN: n = 189 cells from 7. Error bars show SEM.

Regarding Reviewer #2’s Q2, for each trial, the entire 10µL water reward was delivered once. However, we noticed that sometimes the water droplet disperses to the sides of the lick port, therefore, it is hard to determine how many licks it will take the mouse to finish the water. Since it varies trial-to-trial, it is difficult to determine the exact reward duration for each trial. The example trace in Figure 2C only reflects that particular trial because in other licking examples in Figure 4B and 5B, mice all showed sustained licking after water delivery on Day 7.

4. In Activity Analysis and Tuning Coefficients Calculation, the authors performed a resampling of mice with replacement, and the size of the random sample equals that of the original set of mice. Please clarify why this is done this way. Can the authors simply take all mice into analysis?

In our understanding, it is typical to perform the bootstrap analysis with random sample replacement. Based on two classic papers (Efron, *The Annals of Statistics*, 1979; Sitter, *Comparing three bootstrap methods for survey data*, 1992), the authors’ described the advantage of sampling with replacement to obtain a more accurate and unbiased estimate of variance.

Following the reviewer’s suggestion, we repeated the analysis in Figure 3 without sample replacement for all cell types. Interestingly, we noticed that after 1,000 repetitions, the distributions using ‘without replacement’ were much narrower compared to ‘with replacement’ (Author response image 1). This is in line with the ‘without replacement’ method providing an inherently biased estimate of variance that does not capture the true distribution of the data.

5. Both Reviewer #2 and Reviewer #3 pointed out the importance of plotting calcium signals over days without resorting. The authors conclude that "PV-INs that began as highly reliable maintained their reliability to the CS, …, while PV-INs that began as low reliability became significantly more reliable." However, Figure 4F only shows how the percentage of neurons in the high- or low-reliability category changes overtraining. To draw this conclusion, the authors need to track individual neurons and compare the same neuron's reliability on d1 and d7.

We thank the reviewers for their suggestion. We performed the suggested analysis focusing on CS-responsive PV-INs and reward-responsive VIP-INs as these two populations showed an increase in reliability after associative learning (Figure 4F and 5F).

We were able to track each neuron and compare the same neuron’s reliability between Day 1 and Day 7. We first plotted an example mouse to show their reliability increases from Day 1 to Day 7. To do this, we began by sorting the neurons based on their Day 1 reliability and showed each individual neuron’s response to every trial within the session. We then tracked the same neuron to Day 7 and again showed each neuron’s response to every trial while maintaining the same order as Day 1 (without re-sorting on Day 7). We can see that many of the low reliability neurons on Day 1 had more active trials on Day 7 (Figure 4H and 5H).

To quantify that the increase in reliability of individual neurons is significant, we isolated the ‘Low Reliability’ neurons on Day 1 and calculated the change in reliability (Δreliability) for each individual neuron between Day 1 and Day 7. As a control, we randomly sampled the Day 7 session irrespective of the behavioural task and calculated a reliability value. We then subtracted that value from the actual Day 1 reliability value to generate a random Δreliability value for each neuron. By comparing the two distributions, we found the Δreliability from Day 1 to Day 7 in both PV-INs’ responses to CS and VIP-INs’ responses to reward were significantly greater than the random Δreliability in control (Figure 4I and 5I). Together, by tracking the same neurons from Day 1 to Day 7, we showed that the increase in reliability also occurs at the individual neuron level. We have included these results in the manuscript (Line 218 – 234, 271 – 278).

Reviewer #1 (Recommendations for the authors):I would encourage the authors to examine the motor-related activity in their data set, to help shed light on the following questions. Do the cue and reward related changes really reflect local circuit changes as the authors seem to suggest? Or could they potentially reflect changes outside of M1 that are then inherited and readout by M1 cell type? If local circuit changes are occur, one might expect to see changes in the conjunctive responses of cue, reward, and motor activity within individual cells. Changes in network activity between cue, reward, and motor cells may also be observed. It should be possible and worth examining these relationships to tease apart potential mechanisms and impact of non-motor changes in M1.

We thank the reviewer for bringing up this important issue. Please find a detailed response above in ‘Essential Revision #2’. In brief, we identified lick cells in all the cell types and further examined their responses to CS, reward, or CS + reward (Figure 7 —figure supplement 1). We found the majority of active neurons (in all cell types) were non-lick neurons, and the percentage remained stable from Day 1 to Day 7. Moreover, among the lick cells that showed responses to CS, reward, or CS + reward, the percentage of cells in each category also did not change significantly after associative learning. Our working hypothesis is that the changes in CS and reward representations observed in M1 are happening outside of M1, and long-range inputs from other brain regions to a specific cell type in M1 (CS for PV-INs or reward for VIP-INs) are increased and/or strengthened after associative learning. This hypothesis is supported by our control experiments, in which in the no-reward paradigm, PV-INs in M1 did not respond to auditory tone when no water rewards were given. Also, while VIP-INs consistently responded to water rewards on both day 1 and day 7 in the non-paired behavioral paradigm, their reliability did not change when the animals did not learn the association. Future work will involve identifying the brain regions that provide these specific inputs to M1; however, we think that is beyond the scope of this manuscript. We have included a brief discussion of our working hypothesis in the Discussion section (Line 412 – 417).

Reviewer #2 (Recommendations for the authors):1. It is helpful to provide further details about the genetic background of the transgenic animals. Are they homozygous or heterozygous? What genetic background are they maintained in? Also, it would be helpful to indicate the number of neurons imaged per mouse.

Following the reviewer’s suggestion, we have included the new information regarding the transgenic mice in the Methods (Line 709 – 710), and an additional table with the number of neurons imaged per mouse (Supplementary File 1). The figure legends indicate the total number of ROIs across all mice for each experimental condition, and the table provides further details including the number of active cells on day 1 (irrespective of the behavioral task), and the number of active cells tracked from day 1 to day 7 from each individual mouse for all experimental conditions used.

2. A question about the behavioral setup. How long is the water reward presented? How many licks does it take the mouse to consume the 10 ul water reward? It seems from Figure 1B that most licks occur when water is no longer available. Also, the example in Figure 2C suggests that post-cue licking is far fewer on d7 than on d1, suggesting that the mouse has learned to inhibit non-rewarded licking. This seems not to agree with Figure 1C.

We thank the reviewer for the question. The 10µL water reward was delivered all at once. However, we noticed that sometimes the water droplet gets dispersed to the sides of the lick port, and the mouse continues to lick although no additional water was delivered. Hence, it is hard to determine how many licks it takes for the mouse to finish the water. Since it varies trialto-trial, it is difficult to determine the exact reward duration for each trial. The example in Figure 2C only reflects that particular trial because in other licking examples provided in Figure 1C, 4B, 5B, mice all showed continuous licking after water delivery on Day 7.

3. As imaging is conducted in M1, how is the response of the cells related to licking? Can the authors make a licking-triggered average of Ca traces for comparison?

We thank the reviewer for bringing up this important issue. Please find a detailed response above in ‘Essential Revision #2’. In brief, we did not find many lick neurons (from all cell types) present in M1, and the percentage of lick and non-lick neurons did not change after associative learning. This is consistent with previous optical and electrical micro-stimulation studies that showed tongue, jaw, and lip are more reliably evoked in the Anterior-Lateral-Motor area (ALM) (PMID: 20376005, 2036613), rather than M1. We have also included example individual ca^2+^ traces during ITI lick bouts for each of the cell types (Figure 2 —figure supplement 1).

4. In Figure 2B: is the scale bar 10% or 10, i.e., 1000%? Many transients show a very slow onset, which is not consistent with the rapid rising phase of GCaMP6f signals as shown in many previous publications. Also, previous publications show that PV and SOM interneurons have very synchronized activity in mPFC (Pinto and Dan, 2015 Neuron) and secondary motor cortex (Garcia-Junco-Clemente et al., 2019 Cell Report). Is it true in M1? Can the authors give some examples of Ca traces of each type of the interneurons? If Ca transients in M1 interneurons exhibit different kinetics from those in pyramidal neurons, how would that affect the choice of analysis criteria?

We thank the reviewer for the question. The ‘10’ in Figure 2B is the z-score value, which means 10 standard deviations from the mean of the neuron’s entire trace.

Following the reviewer’s suggestion, we have provided ca^2+^ traces of each cell type (PN, PV-INs, VIP-INs, SOM-INs). We showed 5 cells during one lick bout and during one trial on Day 1. We then tracked the same cells to Day 7 and showed the ca^2+^ trace of the same neurons during one trial on Day 7. In general, we did not notice much difference in the kinetics between cell types during ca^2+^ imaging (Figure 2 —figure supplement 1). We also assessed published work from other groups using GCaMP6f for in vivo Ca imaging in inhibitory neurons to further verify the kinetics we observed. When we compared the kinetics of our calcium traces from PN, PV-IN, VIP-IN and SOM-IN to those featured in Pinto and Dan, *Neuron*, 2015 (PMID: 26143660), Puscian et al., *Cell Reports*, 2020 (PMID: 32726633) and Poort et al., *Neuron*, 2021 (PMID: 34906356), we did not observe a notable difference. In addition, since we utilized a threshold to determine active events, the rise kinetics should not influence the quantification of active events. We employed a similar active event criterion as Kato et al., *Neuron*, 2015 (PMID: 26586181), where they used GCaMP6s for ca^2+^ imaging in PV-INs and SOM-INs in the auditory cortex and a threshold of over 1.0 z-score for 3 consecutive frames to identify active events. In our paper, we used GCaMP6f (which has faster kinetics than GCaMP6s), and we used a longer criterion for active events (above 1.0 z-score for 5 consecutive frames) than theirs. Hence, we do not believe that the kinetics of different neuron types will affect our analyses.

In regard to synchrony, we generated similar visualization plots as Pinto and Dan, 2015 and Garcia-Junco-Clemente et al., 2019 following the reviewer’s question. In Pinto and Dan, 2015, the authors show ca^2+^ traces from different PV-IN and SOM-IN neurons in mPFC are highly correlated. We qualitatively compared these to our own individual traces (Figure 2 —figure supplement 1) and did not observe synchrony. In Garcia-Junco-Clemente et al., 2019, the authors used heatmaps to visualize PV-IN population activity in M2 over extended periods of time. They found synchrony within two PV-IN subpopulations – one group that was highly active when the mouse was running, and another group that was highly active during stationary periods. Therefore, we performed a similar analysis and used heatmaps to visualize the activity of all PV-Ins within each mouse over 1,000s periods (similar time course as in Garcia-JuncoClemente et al.,); however, we did not observe any synchrony. We also did not observe any synchrony in SOM-INs (Author response image 2). It is possible that synchrony is specific to certain brain regions, specific tasks and/or specific behavioural states.

**Author response image 2. sa2fig2:** (A-B) Examining if PV-INs or SOM-INs in M1 showed synchronized activity. Z-scored activity of all tracked active neurons from an example PV-Cre mouse and SOM-Cre mouse during the first 1,000s of the Day 1 session (left) and the Day 7 session (right). Neuron order along the y-axis was not sorted and was maintained across days. Red arrows indicate trial start time, and white dashed lines indicate water reward delivery. Licking activity is shown above. Different from Garcia-Junco-Clemente et al., 2019, synchrony was not apparent within the PV-IN or SOM-IN population.

5. The 2.5s period chosen for analysis covers both tone presentation and delay. Will the conclusions in Figure 2E-H change if the analysis is restricted to the 1s of tone presentation? In addition, it seems better to use the actual duration of reward presentation (see Q2) for analysis. As the statistical analysis is done by random shuffling, there seems no need to match the period for tone and reward analyses.

We thank the reviewer for the suggestion. Please find a detailed response above in ‘Essential Revision #3’. In brief, we calculated the percent of CS-responsive neurons for each cell type on Day 1 and Day 7 using the 1-sec time window following CS presentation. We found the percentage of CS-responsive neurons decreased significantly compared to the 2.5-sec window in all the cell types, and the percentages were all around 5% of total active neurons (Author response image 1). Since all cell types had a low percentage of CS-responsive cells, we do not think using a 1-sec time window truly represents the CS responses in M1.

6. In Activity Analysis and Tuning Coefficients Calculation, the authors performed a resampling of mice with replacement, and the size of the random sample equals that of the original set of mice. What is the reason to do this? Can the authors simply take all mice into analysis?

We thank the reviewer for the suggestion. Please find a detailed response above in ‘Essential Revision #4’.

7. Figure 3A-B: a little clarification is needed. (1) Is each trace a single trial or the average over trials? (2) What does it mean that "each trace is from the same neuron on d7?" Are the five traces from one neuron, or five neurons? If they are from five neurons, are these the same five neurons in A and B? (3) Why does the color of a single trace change with time?

We apologize for the lack of clarity and confusion. Each trace is the mean z-score from all the trials of one neuron, and they are different neurons on Day 1 and Day 7. We have removed the sentence ‘each trace is from the same neuron on d7’ from the figure legend. The lighter color part of the trace during non-CS or non-reward period was meant to highlight the actual response period. We have now re-made the figures, and the entire trace is a uniform color that represents the correlation value during CS or reward period.

8. The authors conclude that "PV-INs that began as highly reliable maintained their reliability to the CS, …, while PV-INs that began as low reliability became significantly more reliable." However, Figure 4F only shows how the percentage of neurons in the high- or low-reliability category changes over training. The authors need to track individual neurons and compare the same neuron's reliability on d1 and d7 to draw this conclusion. The same issue applies to the analysis of all cell types.

We thank the reviewer for the comments. Please find a detailed response above in ‘Essential Revision #5’. In brief, we tracked each PV-INs in the ‘Low Reliability’ to CS group and each VIP-INs in the ‘Low Reliability’ to reward group from Day 1 to Day 7, and we calculated the change in reliability (ΔReliability) between both days. For the control, we randomly sampled Day 7 irrespective of the behavioral task and generated a reliability value. We then subtracted that value from the actual Day 1 reliability to generate a random ‘ΔReliability’ value for each neuron. By comparing the two distributions, we found the increase in reliability from Day 1 to Day 7 in both Low Reliability PV-INs to CS and Low Reliability VIP-INs to reward were significantly higher than random.

Reviewer #3 (Recommendations for the authors):First of all, the work is great. Analyzing calcium activity from each interneuron cell type is quite difficult, but authors elegantly performed the experiments. I wonder whether you can plot calcium signals over days (day 1 to day 7) without resorting. I understand it is very difficult, but if you have a good imaging quality enough to trace calcium activity from the same cells over several days, it would be nicer to show.

We thank the reviewer for the generous comments and suggestions. We included calcium traces from 5 example neurons for each cell type on Day 1 during one lick bout and during one trial. We then showed the same neurons again on Day 7 during another trial (Figure 2 —figure supplement 1). However, the limitation of showing calcium traces is that we can only visualize one trial at a time. In order to demonstrate that PV-INs and VIP-INs increase their response reliability to CS and reward, respectively, we decided to show every single trial from all neurons from one example mouse. To do that, we first sorted the neurons (of one mouse) based on their Day 1 reliability, and we tracked the same neurons to Day 7 without re-sorting and showed whether each trial was active or not (Figure 4H and 5H). We can clearly see that the ‘Low Reliability’ PV- or VIP-INs from Day 1 showed more active trials on Day 7 after associative learning (Figure 4I and 5I).

Another concern is the lack of control experiments that show no such changes presented in training groups. Otherwise, it is quite good.

We thank the reviewer for the comments. Please find a detailed response above in ‘Essential Revision #1’. In brief, we altered the reward contingencies in the behavioral task and performed control experiments in three separate cohorts of mice. First, we found that tone presentations in the absence of water rewards did not activate PV-INs in M1, as the percent of active PV-INs during the tone response period was not different from that at baseline. In another cohort of mice, we presented the tone with non-paired, randomly-timed water rewards. Intriguingly, PVINs were significantly responsive to the tone on Day 1 but not Day 7, suggesting PV-INs in M1 do not have pure sensory responses to the tone, but instead, they respond to reward-predicting cues. In contrast, VIP-INs remained significantly responsive to randomly timed water rewards, but the response reliability of the VIP-INs in the ‘Low Reliability’ to reward group did not change from Day 1 to Day 7. Together, our results suggest that the increase in reliability in PV-INs to the CS and VIP-INs to reward are specific to associative learning.